



# Disparities in particulate matter (PM$_{10}$) origins and oxidative potential at a city-scale (Grenoble, France) - Part II: Sources of PM$_{10}$ oxidative potential using multiple linear regression analysis and the predictive applicability of multilayer perceptron neural network analysis

Lucille Joanna S. Borlaza[*1], Samuël Weber[1], Jean-Luc Jaffrezo[1], Stephan Houdier[1], Rémy Slama[2], Camille Rieux[3], Alexandre Albinet[4], Steve Micallef[3], Cécile Trébuchon[3], and Gaëlle Uzu[*1]

[1]University of Grenoble Alpes, CNRS, IRD, INP-G, IGE (UMR 5001), F-38000 Grenoble, France
[2]University of Grenoble Alpes, Inserm, CNRS, IAB (Institute of Advanced Biosciences), Team of Environmental Epidemiology applied to Reproduction and Respiratory Health, Grenoble, France
[3]Atmo AuRA, F-38400 Grenoble, France
[4]INERIS, Parc Technologique Alata, BP 2, 60550 Verneuil-en-Halatte, France

*Correspondence to*: LJS Borlaza (lucille-joanna.borlaza@univ-grenoble-alpes.fr) and G Uzu (gaelle.uzu@ird.fr)

**Abstract.** The oxidative potential (OP) of particulate matter (PM) quantifies PM capability to cause anti-oxidant imbalance. Due to the wide range and complex mixture of species in particulates, little is known on the pollution sources most strongly contributing to OP. A one-year sampling of PM$_{10}$ (particles with an aerodynamic diameter below 10) was performed over different sites in a medium-sized city (Grenoble, France). An enhanced fine-scale apportionment of PM$_{10}$ sources, based on the chemical composition, was performed using Positive Matrix Factorization (PMF) method and reported in a companion paper (Borlaza et al., 2020). OP was assessed as the ability of PM$_{10}$ to generate reactive oxygen species (ROS) using three different acellular assays: Dithiothreitol (DTT), Ascorbic acid (AA), and 2,7-dichlorofluorescein (DCFH) assays. Using multiple linear regression (MLR), the OP contribution of the sources identified by PMF were estimated. Conversely, since atmospheric processes are usually non-linear in nature, artificial neural network (ANN) techniques, which employs non-linear models, could further improve estimates. Hence, the multilayer perceptron analysis (MLP), an ANN-based model, was additionally used to model OP based on PMF-resolved sources as well. This study presents the spatiotemporal variabilities of OP activity with influences by season-specific sources, site typology and specific local features, and assay sensitivity. Overall, both MLR and MLP effectively captured the evolution of OP. The primary traffic and biomass burning sources were the strongest drivers of OP in the Grenoble basin. There is also a clear redistribution of source-specific impacts when using OP instead of mass concentration, underlining the importance of PM redox activity over mass concentration. Finally, the MLP generally offered improvements in OP prediction especially for sites where synergistic and/or antagonistic effects between sources are prominent, supporting the value of using ANN-based models to account for the non-linear dynamics behind the atmospheric processes affecting OP of PM$_{10}$.





# 1 Introduction

One of the most critical pollutants in the atmosphere is particulate matter (PM), especially in urban areas that are heavily
impacted by anthropogenic emissions (David et al., 2019; Qiao et al., 2018; Schwela, 2000). Recent studies showed increasing
interest in PM at a city-level allowing assessment of fine-scale pollution variability (Boppana et al., 2019; Dionisio et al., 2010;
Etyemezian et al., 2005; Krasnov et al., 2016; Padhi and Padhy, 2008). The intricate topography and seasonality of particulate
air pollution in the city of Grenoble (France) makes it an ideal location to explore both the small- and large-scale variabilities
of PM pollution accounting for local variations in different urban environments (Calas et al., 2019; Favez et al., 2010;
Srivastava et al., 2018; Tomaz et al., 2016, 2017; Weber et al., 2019). Such small-scale variabilities for mass and chemical
composition have been recently addressed in a companion paper (Borlaza et al., 2020).

Many research studies have focused on the links between PM mass exposure and various adverse health effects (Dabass et al.,
2018; Delfino et al., 2005; Du et al., 2016; Hime et al., 2018; Lao et al., 2019; Matus C. and Oyarzún G., 2019; Pope et al.,
2009; Pope III, 2002; Winterbottom et al., 2018). However, it is also of high concern to improve the understanding of the PM
sources in relation with such health impacts. Indeed, oxidative stress is now well recognized as one of the main biological
mechanisms considered to be contributing to these detrimental impacts from air pollution exposure through the capability of
PM to generate reactive oxygen species (ROS) within the lung, which leads to pro-inflammatory responses that can ultimately
result in apoptosis (Ayres et al., 2008; Baulig et al., 2003; Dhalla et al., 2000; Donaldson et al., 2001; Jin et al., 2018; Kelly,
2003; Leni et al., 2020; Mudway et al., 2020; Nel, 2005; Piao et al., 2018). The oxidative potential (OP) of PM, with its ability
to integrate the effects of several characteristics of PM, including chemical composition, and its capability to account for anti-
oxidant imbalance into a consolidated measure, makes an interesting complementary to regulated metrics of ambient PM
exposure (Bates et al., 2019; Daellenbach et al., 2020; Guo et al., 2020; Gurgueira et al., 2002; Park et al., 2018; Shiraiwa et
al., 2017; Valavanidis et al., 2008).

Most studies often correlate OP from PM with chemical species in ambient aerosols (Bell and HEI Health Review Committee,
2012; Boogaard et al., 2012; Borlaza et al., 2018; Cassee et al., 2013; Janssen et al., 2014; Perrone et al., 2016; Pietrogrande
et al., 2018; Rohr and Wyzga, 2012; Yang et al., 2015). However, due to the wide range and complex mixture of PM and the
dynamic atmospheric processes to consider, the main drivers of OP can be difficult to highlight (Calas et al., 2019). Several
methods have been used to assign the sources of OP, including the application of receptor modelling techniques such as
Positive Matrix Factorization (PMF) and Chemical Mass Balance (CMB) (Ayres et al., 2008; Bates et al., 2015; Cesari et al.,
2019; Fang et al., 2016; Paraskevopoulou et al., 2019; Verma et al., 2014; Weber et al., 2018, 2021; Yu et al., 2019; Zhou et
al., 2019), Principal Component Analysis (PCA) (Borlaza et al., 2018; Conte et al., 2017), and Robotic Chemical Mass Balance
(RCMB) coupled with Multiple Linear Regression (MLR) analysis (Argyropoulos et al., 2016). With these current techniques,
the OP of PM has been linked to specific emission sources and their estimated contributions. However, because numerous
factors could affect OP and the non-linear relationship of redox active components of PM is generally observed (Arangio et





al., 2016; Calas et al., 2017; Charrier and Anastasio, 2015; Li et al., 2012; Xiong et al., 2017; Yu et al., 2018), the traditional deterministic models could be, in some way, limited.

Approaches using artificial neural network (ANN) analysis have demonstrated enhanced results compared to classical models when predicting PM from different variables such as meteorological data (Abderrahim et al., 2016; Chaloulakou et al., 2003; Díaz-Robles et al., 2008; Hooyberghs et al., 2005; Huang and Kuo, 2018; McKendry, 2002; Papanastasiou et al., 2007; Perez

and Reyes, 2006), satellite-derived aerosol products (Gupta and Christopher, 2009), and other traffic-related variables (Cabaneros et al., 2020, 2017; Gietl and Klemm, 2009; He et al., 2015). The ANN-based models, such as multilayer perceptron (MLP), support pattern recognition and could extract trends from non-linear data, making it an interesting and competitive innovative method of analysis in many scientific disciplines, including air quality studies (Cabaneros et al., 2019; Chattopadhyay and Bandyopadhyay, 2007; Dorling et al., 2003; García Nieto et al., 2018; Gupta and Christopher, 2009; Jiang

et al., 2004; Ordieres et al., 2005; Perez and Reyes, 2006). Since atmospheric processes are generally non-linear in nature, exploring the features of MLP could provide meaningful results closer to realistic estimates than most linear models (Elangasinghe et al., 2014; Eldakhly et al., 2017; Gerken et al., 2006; Kukkonen, 2003; Nathan et al., 2017; Rahimi, 2017).

This study takes advantage of the enhanced source apportionment obtained in the companion paper (Borlaza et al., 2020), revealing the fine-scale spatiotemporal characteristics of PM sources within a medium-size city area (Grenoble basin),

specifically in three different urban environments (background, hyper-center, and peri-urban typologies). Here, the main drivers of OP are first attributed to PM sources (resolved by PMF) using a classical MLR analysis. Second, the possible advantages of MLP analysis are also evaluated to compare MLP prediction of OP activity with MLR prediction. In summary, by taking the opportunity of this unique database on PM chemistry and OP, we aim to investigate mainly on two innovative questions:

1. Is there variability in the OP activity within a medium-sized urban area, and can this be related to the variability of the contributions of the emissions sources?

2. Can MLP be used to accurately model the spatiotemporal evolution of OP by taking the PM source contributions as input variables and if so, does it catch the non-linear pattern of OP?

**2 Materials and methods**

**2.1 Site description and PM$_{10}$ sampling collection**

The sampling sites and samples used in this study are described in detail in the companion paper (Borlaza et al., 2020). Briefly, the sampling sites are located in the city of Grenoble in the southeast of France, as illustrated in Figure 1. The mountainous environment in the area restricts atmospheric movements and promotes the development of atmospheric thermal inversions, resulting in an increase of pollutant concentrations, especially during the winter season (Bessagnet et al., 2020; Tomaz et al.,

2017). The three measurement sites are located in an urban background (UB, Les Frênes), urban hyper-center (UH, Caserne de Bonne), and peri-urban (PU, Vif), all within 15 km from the city center of Grenoble. The UB site is an established urban



background reference site for the regional air quality monitoring network (Atmo Auvergne Rhône-Alpes) in the south of the city and largely investigated previously (Srivastava et al., 2018; Tomaz et al., 2016). The PU site is in a suburban area having rural residential areas adjacent to an urbanization (low-density area), where biogenic emissions are prominently expected as the site is on the foot of the Vercors and Belledone mountain ranges. Lastly, the UH site is in the hyper-center of Grenoble and, despite being in a pedestrian area, is the most highly exposed to surrounding commercial and traffic emissions amongst the three sites.


The daily (24-h) filter-based $PM_{10}$ (particles ≤10 μm in diameter) sampling was performed with a 3-day interval for about one year (February 28, 2017 to March 10, 2018, sampling starts at 00:00 CEST) obtaining a total of about 130 samples per site.

$PM_{10}$ was collected using a high volume sampler (Digitel DA-80, 30 $m^3$ $h^{-1}$) onto 150 mm-diameter quartz fiber filters (Tissuquartz PALL QAT-UP 2500 diameter 150 mm) following the recommendations of EN 12341:2014 procedures (CEN, 2014). All filters underwent a preheating treatment at 500°C for 12 hours to avoid any organic contamination. Additionally, field blank filters (n=20) were collected to determine the detection limits of the applied chemical analysis and to secure quality of samples during transport, setup, and recovery. The total $PM_{10}$ mass concentration was also simultaneously measured using

tapered element oscillating microbalance equipped with filter dynamics measurement systems (TEOM-FDMS) (CEN, 2017; Grover, 2005).

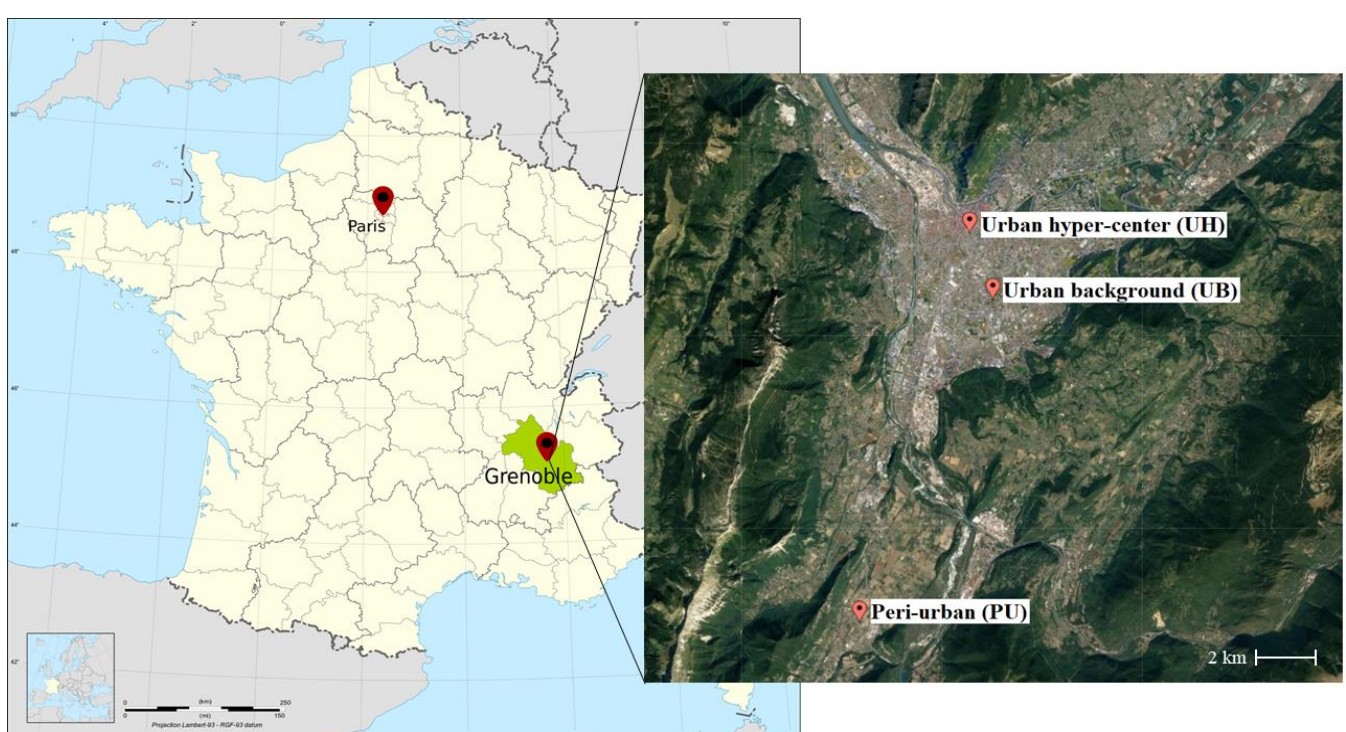

**Figure 1: Study area in Grenoble (France) on a European map (left) and location of the three urban sites (right), namely Les Frênes**
**or UB (urban reference background site), Caserne de Bonne or UH (urban hyper-center site), and Vif or PU (peri-urban site).**
**©OpenStreetMap contributors 2020. Distributed under a Creative Commons BY-SA License**



## 2.2 Chemical characterization

All samples were subjected to several chemical analyses to quantify major and minor constituents of $PM_{10}$ including organic carbon (OC), elemental carbon (EC), ions (sodium ($Na^+$), ammonium ($NH_4^+$), potassium ($K^+$), magnesium ($Mg^{2+}$), calcium

($Ca^{2+}$), chloride ($Cl^-$), nitrate ($NO_3^-$), sulfate ($SO_4^{2-}$)), methane sulfonic acid (MSA), organic acids (3-MBTCA, pinic acid, phthalic acid), anhydro-sugars (levoglucosan and mannosan) and primary saccharides (arabitol and mannitol, hereafter summed up and referred as polyols), cellulose, and elements (Al, As, Ba, Cd, Cr, Cu, Fe, Mn, Mo, Ni, Pb, Rb, Sb, Se, Sn, Ti, V, Zn). Detailed descriptions of the chemical analyses are available in the companion paper (Borlaza et al., 2020) and a summary of $PM_{10}$ characteristics is available in Table S1 the supplementary information (S1).

## 2.3 OP analysis

For OP analysis, the filters were subjected to $PM_{10}$ extraction using a simulated lung fluid (SLF) solution composed of a Gamble + DPPC (dipalmitoylphosphatidylcholine) mixture (Calas et al., 2018). In order to maintain a constant amount of extracted $PM_{10}$, filter punches were adjusted by area to obtain iso-mass at 25 µg ml⁻¹. No filtration was done in order to include both water soluble and insoluble particles. Such extraction method has been adopted to facilitate the extraction of $PM_{10}$ in

conditions closer to lung physiology (Calas et al., 2017). The OP activity can be represented using two different measures: 1) the mass-normalized OP activity ($OP_m$), where OP is normalized by the mass of $PM_{10}$ (µg), and 2) the volume-normalized OP activity ($OP_v$), where OP is normalized by the sampled air volume (m³). The $OP_m$ is the intrinsic OP property of one µg of PM, while $OP_v$ represents the PM-derived OP exposure. Three a-cellular complementary assays were used to perform OP measurements and are briefly described in the following sections. All samples were subjected to triplicate analysis and each

sample results in the mean of such triplicate. The common coefficient variation (%CV) is between 0 and 10% for each assay.

### 2.3.1 Dithiothreitol (DTT) assay

DTT is considered as a chemical surrogate to cellular reducing agents, nicotinamide adenine dinucleotide (NADH) and nicotinamide adenine dinucleotide phosphate-oxidase (NADPH), to mimic in vivo interactions of PM and biological oxidants. The consumption of DTT in the assay is inferred as a measure of the ability of the PM to transfer electrons from DTT to oxygen

thereby producing reactive oxygen species (ROS). Our procedure is based on a modified protocol by Cho et al. (2005), as described in Calas et al. (2018). The $PM_{10}$ extracts were reacted with DTT resulting to the consumption of DTT in the solution. The remaining DTT is then titrated with 5,5-dithiobis-(2-nitrobenzoic acid) (DTNB) to produce a yellow chromophore (5-mercapto-2-nitrobenzoic acid or TNB), which is in direct proportion to the amount of reduced DTT remaining in solution after the reaction with the $PM_{10}$ extract. These mixtures were injected in a 96-well plate (CELLSTAR, Greiner-Bio) and the

consumption of DTT (nmol min⁻¹) was determined by following the TNB absorbance at 412 nm wavelength using a microplate-reader (TECAN spectrophotometer Infinite M200 Pro) at 10-minute intervals for a total of 30 minutes of analysis time.





### 2.3.2 Ascorbic acid (AA) assay

The AA assay is based on a modified procedure by (Kelly and Mudway, 2003), as described in Calas et al. (2018), using a respiratory tract lining fluid (RTFL). This assay uses AA, a known antioxidant which prevents the oxidation of lipids and
proteins in the lung lining fluid (Valko et al., 2005). The consumption of AA (nmol min$^{-1}$) in the assay is inferred as the OP of $PM_{10}$ quantified by the transfer of electrons from AA to oxygen ($O_2$). Similar to the DTT assay, the $PM_{10}$ extracts were reacted with AA into a 96-well plate UV-transparent (CELLSTAR, Greiner-Bio). The absorbance was measured at 265 nm using a plate-reader (TECAN spectrophotometer Infinite M200 Pro) at 4-minute intervals for a total of 30 minutes of analysis time.

### 2.3.3 Dichloro-dihydro-fluorescein diacetate (DCFH) assay

The 2,7-dichlorofluorescin (DCFH) assay is commonly used for detecting intracellular $H_2O_2$ and oxidative stress using a non-fluorescent probe through the formation of a fluorescent product (dichlorofluorescein or DCF) in the presence of ROS and horseradish peroxidase (HRP). The DCF is measured by fluorescence at the excitation and emission wavelengths of 485 and 530 nm, respectively, every 2 minutes for a total of 30 minutes of analysis time. The ROS concentration in the sample is calculated in terms of $H_2O_2$ equivalent based on a $H_2O_2$ calibration (100, 200, 300, 400, 500, 1000, and 2000 nmol).

## 2.4 Data analysis

### 2.4.1 Synthesis of the methodology used for $PM_{10}$ source apportionment

The source apportionment performed on this dataset has been described into details in the companion paper (Borlaza et al., 2020). In brief, the PMF methodology used the EPA PMF5.0 software (US EPA, Norris et al. (2014)) and closely follows the parameterization used in previous works by our group (Favez et al., 2017; Waked et al., 2014; Weber et al., 2019, 2021) with
a few relevant modifications.

The input variables used were mass concentration and uncertainty levels of $PM_{10}$ and its chemical composition (a total of 35 variables) including OC, EC, ions, elements, and some organic markers (MSA, levoglucosan, mannosan, polyols, pinic acid, 3-MBTCA, phthalic acid, and cellulose). The associated uncertainties were calculated based on a method proposed by Gianini et al. (2012). Specific geochemical constraints, based on expert prior knowledge, were added to the solution using the ME-2
solver (Paatero, 1999), particularly for the traffic source factor (Charron et al., 2019). The statistical validity of the solution and the uncertainties were estimated using the bootstrap and displacement methods following the European recommendation for source apportionment studies (Belis et al., 2019; Brown et al., 2015). The specific tracers used to identify the sources are presented in Table S2 in the supplementary information (S2).



### 2.4.2 Multiple linear regression (MLR) analysis

A multiple linear regression (MLR) analysis was performed to attribute OP from the PMF-resolved sources of PM$_{10}$, following
the OP deconvolution methodology proposed by Weber et al. (2018). The $OP_v$ from the three assays were used individually as
the dependent variable, while the PMF-resolved source contributions were used as independent variables, as shown in Eq.1:

$$OP_{obs} = (G_n \times \beta_n) + \varepsilon, \tag{1}$$

where OP$_{obs}$ is the observed daily $OP_v$, matrix of size d×1 in nmol$_{reactant}$ min$^{-1}$ m$^{-3}$, G is the contribution of the sources from the

PMF in µg m$^{-3}$ of size d×n and β is the regression coefficient representing the intrinsic OP (or the $OP_m$) of size 1×n in nmol
min$^{-1}$ µg$^{-1}$. Finally, $\varepsilon$ is the residual term accounting for the difference between the observed and modeled OP of size d×1 in
nmol$_{reactant}$ min$^{-1}$ m$^{-3}$. The OP contribution of each source is calculated by multiplying the source-specific regression coefficient
by the contribution of the source to PM$_{10}$ ($G_k \times \beta_k$).

### 2.4.3 Multilayer perceptron (MLP) neural network analysis

#### 2.4.3.1 Background of the MLP analysis

The MLP analysis is designed using a feed forward learning model (Calcagno et al., 2010; García Nieto et al., 2018; Salazar-
Ruiz et al., 2008) that produces a predictive model for one or more output variables ($OP_v$) based on the values of the input
variables (PM$_{10}$ source contributions). The three main components of MLP are: 1) the input layer, 2) the hidden layer, and 3)
the output layer. Generally, the MLP consists of interconnected layers of artificial neurons that form a network using a set of

input data and draws it onto a set of output data, which are then used to further train the neural network through a back-
propagation process (Bishop, 1995; Fontes et al., 2014; Kim and Gilley, 2008). In this study, the neural network architecture
was limited to a one hidden layer design to demonstrate the applicability of non-linear models, even only with a rudimentary
architecture, and to compare its predictive capability against that of MLR.

#### 2.4.3.2 Implementation of the MLP

As an initial step, a rescaling process is applied to both the input and output layers to eliminate potential bias due to the range
of variance within the dataset (Gardner and Dorling, 1998). Each variable is standardized by subtracting the mean observed
value and then divided by the standard deviation. The daily contributions of the PM sources obtained from the PMF were fed
in the input layer to the hidden layer. The MLP analysis was performed for each site using the $OP_v$ from each assay ($OP_v^{DTT}$,
$OP_v^{AA}$, and $OP_v^{DCFH}$) as multiple variables in the output layer (see Figure 2), making a set of 9 independent studies. At each

node (or neuron), the information given by the input neurons are condensed into a unique value and propagated to the next
layer. For instance, the MLP described in Figure 2 is formally defined by Eq. 2 for the first layer (hidden layer):

$$\forall j \in \{1, \dots, l\}, z_j = H\left(\sum_{i=1}^{d} w_{i,j}^G \times x_i + w_{0,j}^G\right) \tag{2}$$





with $w_{i,j}^G$ the weight of the neuron between the input and hidden layer and $w_{0,j}^G$ an activation constant for neuron j. The activation function H is often non-linear.

To sum up, the hidden layer develops the input data and deciphers the relationship of the neurons within the MLP network. The number of neurons in the hidden layer was determined automatically by the estimation algorithm. With the activation function, the hidden layer transfers a response onto the output layer. The activation functions tested in this study were sigmoid and hyperbolic tangent (TanH) as these are appropriate for continuous dependent variables (IBM, 2016). A weight initialization was preset for potential occurrence of vanishing gradients (Bengio et al., 1994; Hochreiter, 1998; Hochreiter and Schmidhuber,

1997). The scaled conjugate and stochastic gradient descent optimization algorithms were tested to obtain the optimal weights in both the input and output layers (Slini et al., 2006; Vakili et al., 2015). The various MLP architectures tested are summarized in the supplementary information (S3).

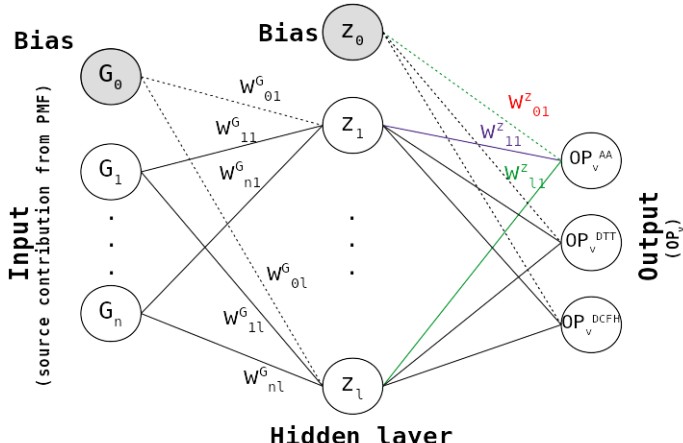

**Figure 2: The MLP neural network architecture used in this study, where *n* refers to the number of source, *G* is the normalized contribution from the PMF, and $OP_v$ is the different volume-normalized OP activities ($OP_v^{DTT}, OP_v^{AA}, and\ OP_v^{DCFH}$)**

The dataset was partitioned into: 1) the training set accounting for 80%, and 2) the testing set accounting for 20% of the dataset. For each of the 9 studies, the training set contains data points that were used to train the MLP, while the testing set is an independent set of data points used to monitor errors during the training step. During the training step, the MLP is continually

developed and refined until the weighting values between the nodes accurately predict the outcome (i.e., minimal possible errors). To prevent the model from over-fitting, a set of stopping rules are applied to terminate the training of the MLP when any of these scenarios occur such as: 1) there is no decrease in prediction error for more than 1 step, 2) the maximum training time is reached (15 minutes), 3) the minimum relative change in training error is reached (0.0001), 4) the minimum relative change in training error ratio is reached (0.001). A maximum of 1000 data passes (epochs) are stored in memory until this step

is completed. Using the results obtained in the training step, the results are validated in the testing step to check the performance of the network by assessing its forecasting capability on data points outside the training set. The MLP neural network analysis was performed using IBM SPSS Statistics for Windows, version 20 (IBM Corp., Armonk, N.Y., USA).





**2.4.3.3 Demonstration of the non-linear behaviour of sources using the MLP models**

Since MLP analysis should account for the interactions between $PM_{10}$ sources, the non-linear atmospheric dynamics causing

possible synergistic or antagonistic effects on the OP activity can be captured. To visualize such possible non-linear behaviour, the MLP models obtained were applied on a set of dummy datasets. Each dummy dataset consists of the same mass contributions (from PMF analysis) of each source (in µg m$^{-3}$) as in the original dataset but setting one source ($n$) to zero. This modelled OP using a dummy dataset ($MLP_n$) is subtracted to the modelled OP by the original MLP model ($MLP$) (containing all source contributions). This difference represents a source-specific OP contribution and their summation

($MLP_{sum}$) is described in Eq. 3:

$$MLP_{sum} = \sum MLP_n \qquad (3)$$

For example, if the biomass burning source contributions was set to zero in the dummy dataset ($MLP_{n=biomass\ burning}$), then ($MLP - MLP_{n=biomass\ burning}$) represents the MLP-modelled OP contribution of the biomass burning source. Assuming there is completely no synergistic or antagonistic effects between $PM_{10}$ sources, then the original MLP-modelled OP contributions

should be equal to the sum of all source-specific OP contributions ($MLP = MLP_{sum}$). In cases where $MLP > MLP_{sum}$, then synergistic effects are highlighted between some $PM_{10}$ sources resulting in an increased MLP-modelled OP activity. Conversely, where $MLP < MLP_{sum}$ highlights antagonistic effects between some $PM_{10}$ sources resulting in a decreased MLP-modelled OP activity.

**2.4.4 Statistical analysis**

For the comparison of temporal variations of the observed measurements, all the correlations were evaluated using Spearman rank correlation coefficients ($r_s$), where $p \leq 0.05$ is considered statistically significant. For the evaluation and comparison of model performance between the MLR and MLP results, a number of performance indicators were calculated such as the goodness-of-fit ($R^2$), root mean square error (RMSE), and Pearson correlation coefficient ($r$). The STATA/SE version 15.1 software (College Station, TX, USA) or Python libraries was used for the statistical analyses.

**3 Results and discussion**

**3.1 Temporal variation of PM$_{10}$ and OP activity**

The daily distributions of $PM_{10}$ and OP activity ($OP_v^{DTT}$, $OP_v^{AA}$, and $OP_v^{DCFH}$) for each site are provided in the supplementary information (S4). Detailed discussion of the temporal variability of $PM_{10}$ sources is available in the companion paper (Borlaza et al., 2020).

Overall, the average $PM_{10}$ concentrations on days of measurements were higher during the colder months (October to April) at 17±10 µg m$^{-3}$ and lower during the warmer months (May to September) at 10±4 µg m$^{-3}$ in the city of Grenoble. With the alpine environment and the atmospheric dynamics in the study area, the occurrence of atmospheric inversions and the restriction of strong winds often results to higher concentration levels of air pollutants especially in the winter season





(Bessagnet et al., 2020; Tomaz et al., 2017). Such observed seasonality in PM$_{10}$ mass concentration is also commonly explained

by higher contributions from the biomass burning source in the colder seasons, especially in an alpine valley as previously reported in previous studies (Calas et al., 2019; Favez et al., 2010; Herich et al., 2014; Srivastava et al., 2018; Tomaz et al., 2016, 2017; Weber et al., 2018, 2019). In the same way, a seasonality is displayed in OP activity in the Grenoble basin as well. In fact, the average daily OP activity levels during the winter season can be up to 2, 7, and 5 times higher than in summer season for $OP_v^{DTT}$, $OP_v^{AA}$, and $OP_v^{DCFH}$, respectively. Indeed, the observed strong seasonality (higher OP during winter, lower

OP during summer) at all sites could induce a high spatial homogeneity between sites as well. However, there are a number of local features observed at different sites such as spikes in the OP activity during the warmer months at UH and PU sites (see Figure S1 in the supplementary information (S5)). These spikes are prominently seen in the $OP_v^{DTT}$, with also some occurrences in the $OP_v^{AA}$ and $OP_v^{DCFH}$, which also emphasizes on the sensitivity of each assay.

Previous studies have reported that the $OP_v^{DTT}$ has shown higher sensitivity with organics, metals, and the synergistic effect of

the two (Bates et al., 2019; Dou et al., 2015; Fang et al., 2017; Gao et al., 2020b, 2020a; Jiang et al., 2019; Weber et al., 2021; Yu et al., 2018), while $OP_v^{AA}$ being sensitive mostly to metals concentrations (Bates et al., 2019; Crobeddu et al., 2017; Visentin et al., 2016; Weber et al., 2021). In our study, a good correlation (r=0.68) was found between $OP_v^{DTT}$ and $OP_v^{AA}$ when all sites are combined (see Figure S1 in the supplementary information (S5)), possibly affected by the local features solely captured by the DTT assay. Due to the sensitivity to various ROS and RNS (reactive nitrogen species) of most molecular probes, the

sensitivity of DCFH assay to specific components of PM$_{10}$ can be difficult to isolate (Bates et al., 2019; Jovanovic et al., 2019). However, $OP_v^{DCFH}$ showed good correlation (r=0.68) with $OP_v^{DTT}$ and an even stronger correlation (r=0.93) with $OP_v^{AA}$ (see Figure S2 in the supplementary information (S5)).

The comparison of the two OP measures, $OP_v$ and $OP_m$, of each OP assay can provide information regarding the dependency of OP activity to PM$_{10}$ mass concentration. As shown in Figure S3 in the supplementary information (S5), there is only a

moderate correlation (r=0.51) between $OP_v^{DTT}$ and $OP_m^{DTT}$ suggesting the dependency of DTT assay to chemical composition rather than PM$_{10}$ mass concentration. On the other hand, both $OP^{AA}$ (r=0.76) and $OP^{DCFH}$ (r=0.70) showed good correlations between their measures per volume or per mass pointing out their dependency to PM$_{10}$ concentrations and, indeed, a potential stronger influence by meteorological conditions, a key driver for concentrations in Alpine valleys.

**3.2 Spatial variation of OP activity**

The seasonal mean ratios (MR) of OP activities between sites are presented in Figure 3, calculated by averaging the daily ratios of volume-normalized OP activities ($OP_v^{DTT}$, $OP_v^{AA}$, and $OP_v^{DCFH}$) between the sites (Hyper-center/Background (UH/UB), Hyper-center/Peri-urban (UH/PU), and Background/Peri-urban (UB/PU)) by season, where winter is from December to February, spring is from March to May, summer is June to August, and autumn is September to November.

Generally, there is spatial homogeneity (MR closer to 1) in OP between the UB and UH sites in line with the findings from

the companion paper (Borlaza et al., 2020). Their similarities in terms of PM$_{10}$ sources has been previously attributed to



similarities in source contribution not only from common sources (e.g., biomass burning and nitrate-rich) but also in terms of specific local sources in these sites such as primary traffic, mineral dust, and, to a lower extent, the industrial factor. This could be attributed not only to their proximity in terms of geographical location, but also by their resemblance in typology resulting to similarities of both $PM_{10}$ and OP variabilities.


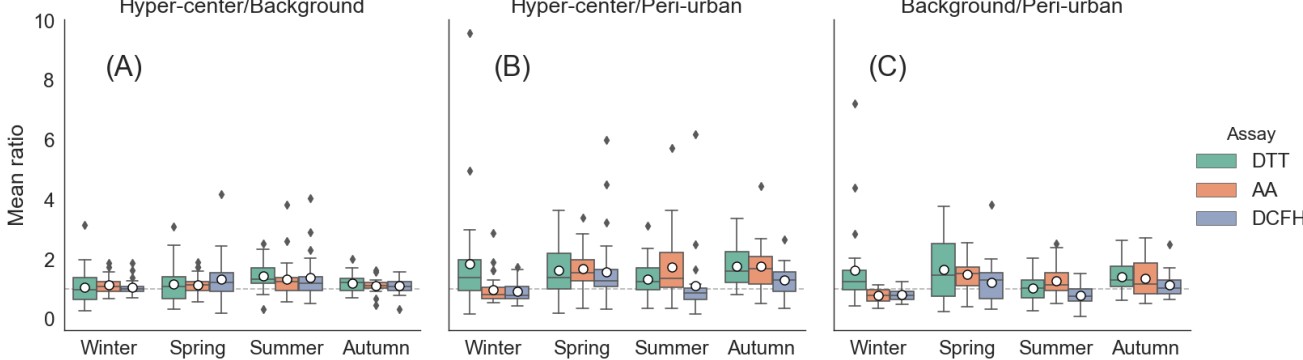

**Figure 3: Seasonal mean ratios (MR) between the sites ((A) Hyper-center/Background (UH/UB), (B) Hyper-center/Peri-urban (UH/PU), and (C) Background/Peri-urban (UB/PU)) using volume-normalized OP activities ($OP_v^{DTT}$, $OP_v^{AA}$, and $OP_v^{DCFH}$). Dashed**
**grey line denotes MR equal to 1 suggesting total spatial homogeneity. Boxplot mean marked by white circle and median marked by black line.**

Conversely, there is an observed variability in the MR in UH/PU and UB/PU suggesting weaker homogeneity (MR farther to 1) in the PU site compared to sites closer to the city-center (UH and UB sites). For example, the PU site can be strongly influenced by some event days with extremely low $OP_v^{DTT}$ especially in the winter season ($OP_v^{DTT}$<0.1 nmol min$^{-1}$ m$^{-3}$, n=3)
resulting to an increase in the MRs against other sites. In fact, the MR for $OP_v^{DTT}$ can be as high as 9.6 and 7.2 during winter for the UH/PU and UB/PU ratio. This can also be seen in the other seasons but more prominent between UH and PU sites. Aside from seasonal influences, there are also some differences between assays as observed in the UH/PU and UB/PU ratio during winter. For instance, the MRs in $OP_v^{DTT}$ is notably much higher than the in $OP_v^{AA}$ and $OP_v^{DCFH}$ further highlighting assay sensitivity.

Although spatial homogeneity was generally observed between the sites, there are local features that must be taken into consideration, as well as seasonal influence and OP assay sensitivity. Overall, there is an observed similarity in the spatiotemporal variabilities of $PM_{10}$ and measured OP activity making it even more interesting to determine which of the $PM_{10}$ sources are driving OP.





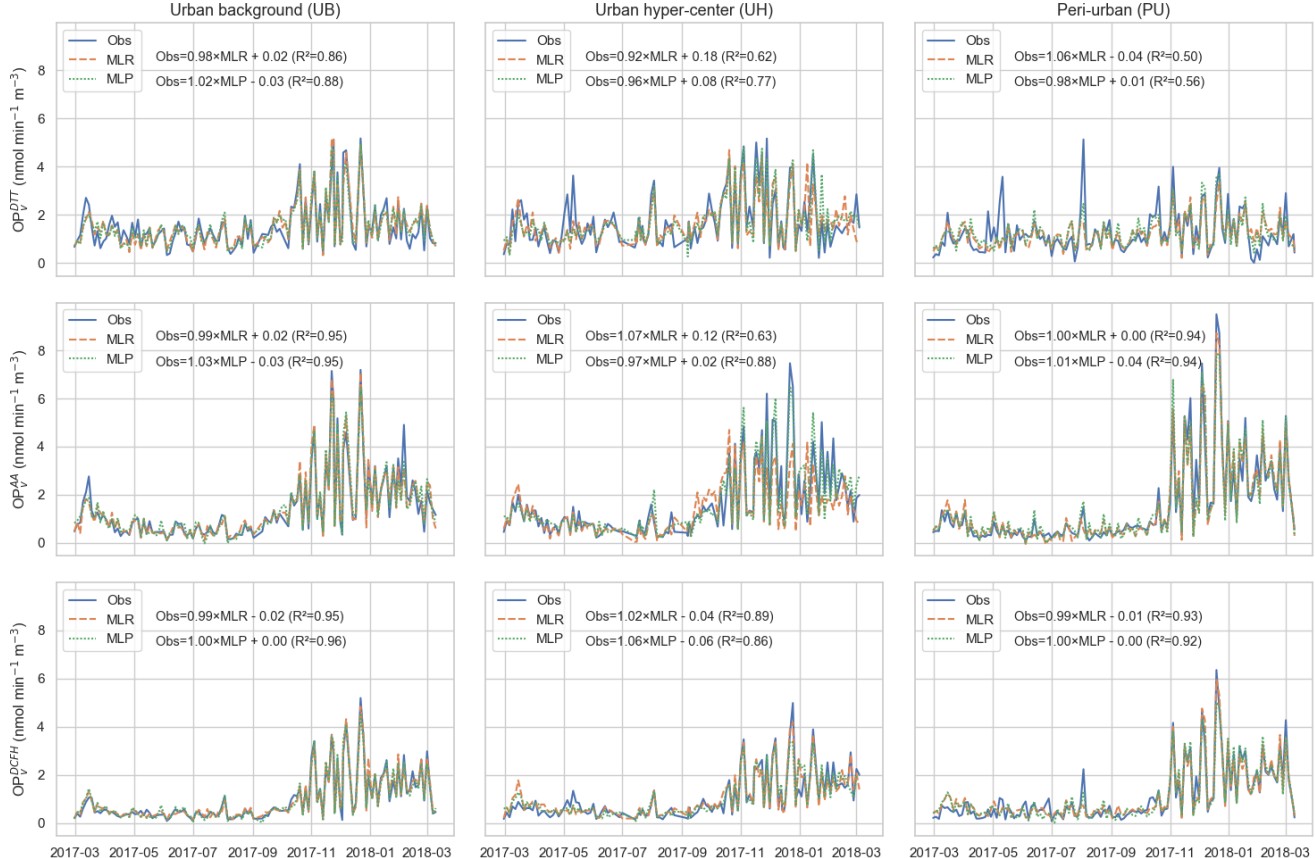

**Figure 4: Comparison of the observed and modelled $OP_v$ ($OP_v^{DTT}$, $OP_v^{AA}$, and $OP_v^{DCFH}$) at different urban sites using MLR and MLP models. The equation of the line and goodness-of-fit ($R^2$) between observed and modelled OP are included.**

### 3.3 Determination of the sources driving OP using multiple linear regression (MLR) analysis

To determine the main drivers of the OP of PM$_{10}$, an OP deconvolution method was performed with a classical MLR analysis
following the proposed method by Weber et al. (2018) using the source contributions obtained in the PMF studies presented
in the companion paper (Borlaza et al., 2020) and the measured OP at each site.

### 3.3.1 Performance of the MLR models

Thanks to the OP deconvolution method, the measured OP has been attributed to the PM$_{10}$ sources allowing the quantification
of contribution of each source to OP. Generally, the MLR-modelled OPs are well within range of the observed OP activity,
even taking into account the low uncertainties of the measurements as presented in Figure 4. However, there are a few local
features (i.e., high OP events) in the observed $OP_v^{DTT}$ during warmer months in the UH and PU sites that were not captured by
the MLR models. There are also some over-estimations during the colder months (specifically around January to February
2018) at the same sites. Yet, these lead to an acceptable goodness-of-fit ($R^2$) for the MLR-modelled $OP_v^{DTT}$ in the UB





$(R^2=0.80)$, UH ($R^2=0.62$), and PU ($R^2=0.50$) sites, compared to the MLR-modelled $OP_v^{AA}$ (UB: $R^2=0.73$, UH: $R^2=0.63$, and

PU: $R^2=0.94$) and $OP_v^{DCFH}$ (UB: $R^2=0.96$, UH: $R^2=0.89$, and PU: $R^2=0.93$). These associations were also confirmed using

Pearson correlations ($r$) as presented in Figure S6 in the supplementary information (S7).

However, there are instances where models, even those with good $R^2$-values, could have a considerable bias and should be

interpreted with caution. For example, the relationship between the observed and MLR-modelled $OP_v^{AA}$ in the UB site has a

slope of 0.9 but an intercept of 0.7, showing significant deviation between model and measured. Additional details on the

correlation between the observed and MLR-modelled OP activity are summarized in the supplementary information (S6).

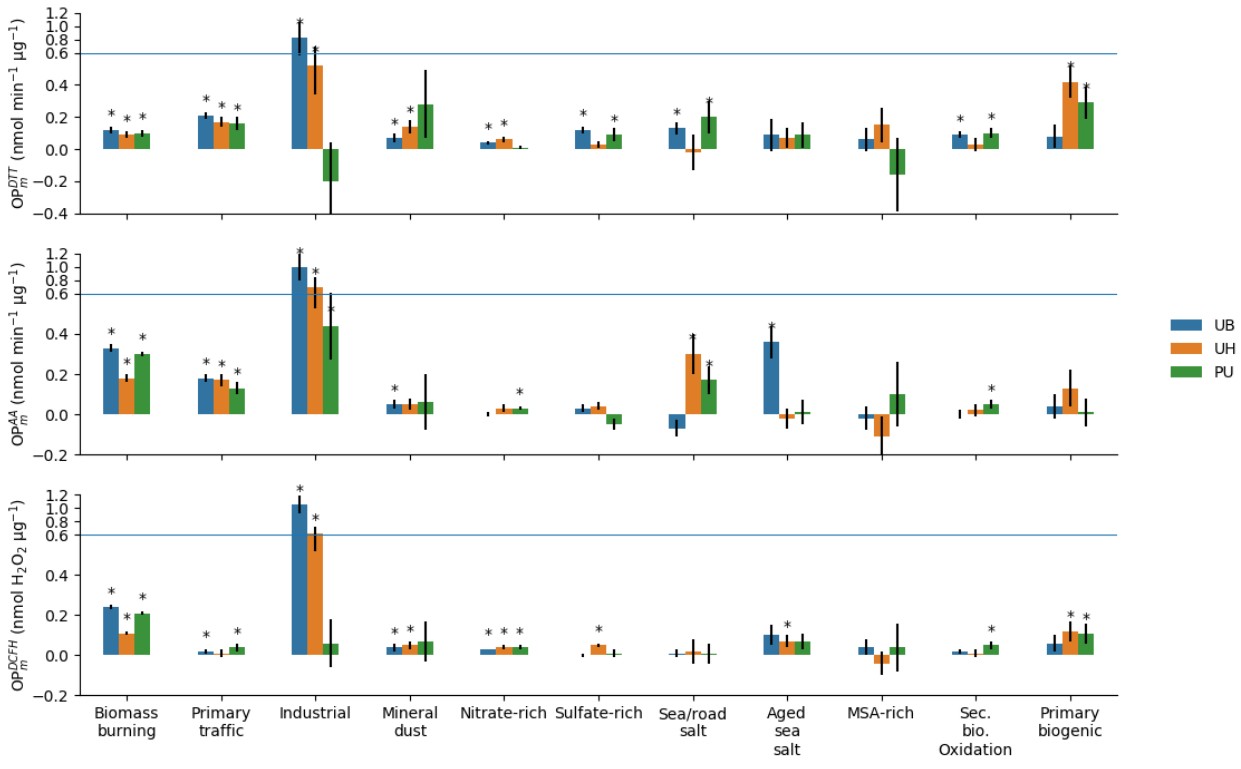

**Figure 5:** Site-specific intrinsic OP ($OP_m$) per source analysis from each assay ($OP_m^{DTT}$, $OP_m^{AA}$, and $OP_m^{DCFH}$) represented by mean (bar) and standard deviation (error bar) based on the MLR (Urban background, UB: blue, Urban hyper-center, UH: orange, Peri-
urban, PU: green). Note: Asterisks represent statistically significant $OP_m$ within 95% confidence interval (p-value≤0.05).

### 3.3.2 Intrinsic OP ($OP_m$) of each PM₁₀ source

The ability of each PM source to induce oxidative stress is represented by the intrinsic OP ($OP_m$) given by the regression

coefficient (β) of the MLR model, as shown in Figure 5. With higher $OP_m$, the source is more redox-active and highly likely

to contribute to the overall OP.





Generally, the statistically dominant sources (based on the MLR models, p-value≤0.05) in every site are the industrial, biomass burning, and primary traffic (except for $OP_m^{DCFH}$ in the UH site) sources, suggesting stronger impact of anthropogenic sources. Both the biomass burning and primary traffic sources have mostly showed significant positive $OP_m$ across all sites. However, amongst the sources with dominant intrinsic OP, it is important to note the variability of the $OP_m$ of the industrial source. Particularly, the industrial source has the highest $OP_m$ for both UB ($OP_m^{DTT}$=0.82±0.24, p≤0.01; $OP_m^{AA}$=0.99±0.20, p≤0.01;

$OP_m^{DCFH}$=1.05±0.13, p≤0.01) and UH ($OP_m^{DTT}$=0.52±0.18, p≤0.01; $OP_m^{AA}$=0.69±0.16, p≤0.01; $OP_m^{DCFH}$=0.62±0.10, p≤0.01) sites. However, for the PU site, the industrial source has a low to negative $OP_m$ for DTT and DCFH assays suggesting that this source has less impact on this specific urban typology. In fact, in the PU site, the highest $OP_m$ was found in different sources, such as the primary biogenic ( $OP_m^{DTT}$ =0.29±0.1, p≤0.01), industrial=0.44±0.17, p≤0.01), and biomass burning ($OP_m^{DCFH}$=0.21±0.01, p≤0.01) sources for DTT, AA, and DCFH assays, respectively.

Although it is clear that anthropogenic sources have higher $OP_m$, there are also impacts from biogenic sources (both primary and secondary biogenic oxidation) that need be considered especially in sites that have an abundance of this type of source. The secondary biogenic oxidation source has only shown statistically significant $OP_m$ in the PU site for all OP assays (also UB site on $OP_m^{DTT}$ only) underlining the influence of site-specific features on $OP_m$.

Aside from biogenic sources, thanks to the enhanced PMF solution used in this study, we were able to determine the redox
characteristics of commonly unresolved sources. The contributions of specific organic tracers (particularly phthalic acid) in some anthropogenic-derived sources, such as sulfate- and nitrate-rich sources, can also point to contributions from anthropogenic secondary organic aerosols (SOA) as discussed in the companion paper (Borlaza et al., 2020). This is particularly important especially that such sources could play a key role in the dynamics of OP of PM$_{10}$ (Daellenbach et al., 2020).

Finally, Weber et al. (2021) discussed the variability of OP at the national scale and the values here are in the ballpark of the national results. A key feature is that the uncertainties of each $OP_m$ can provide information on its statistical significance, therefore offers caution when using these values for modelling purposes.

### 3.3.3 All-sites average OP contribution ($OP_v$) by each PM$_{10}$ source

In terms of overall daily mean contribution, as presented in Figure 6 (see supplementary information (S7) for site-specific
figures), the main contributors of PM$_{10}$ mass are the biomass burning, and the nitrate- and sulfate-rich sources in the Grenoble basin, when taking into account the results from the 3 sites. However, in terms of $OP_v^{DTT}$, the primary traffic source showed the highest contribution (0.33 nmol min$^{-1}$ m$^{-3}$) closely followed by the biomass burning source (0.31 nmol min$^{-1}$ m$^{-3}$). For both $OP_v^{AA}$ and $OP_v^{DCFH}$, the biomass burning source is notably the strongest contributor (0.72 nmol min$^{-1}$ m$^{-3}$ and 0.56 nmol min$^{-1}$ m$^{-3}$, respectively).






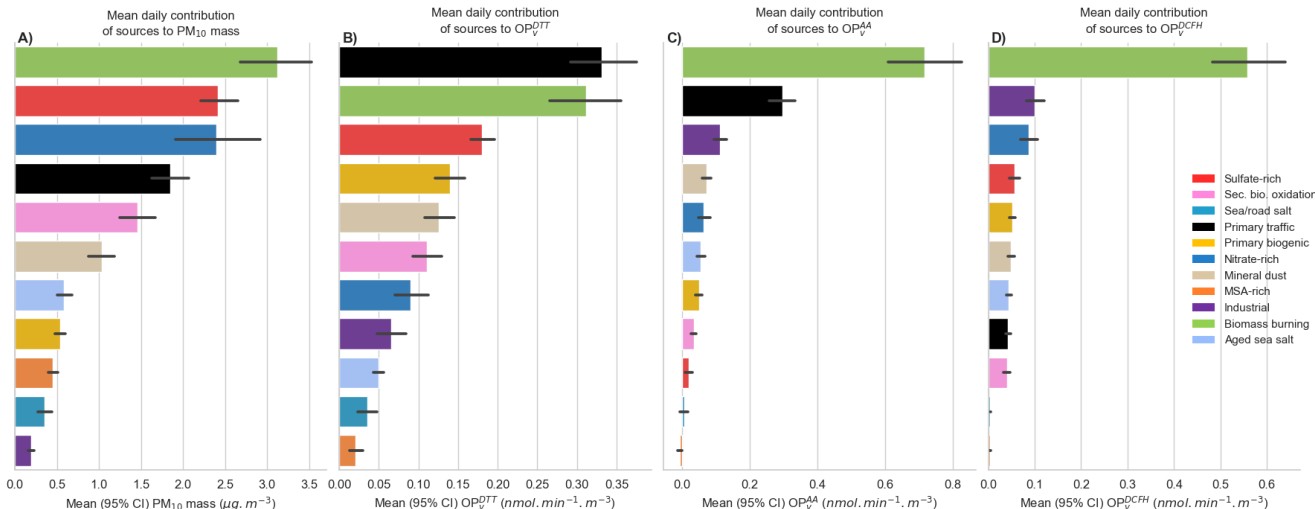

**Figure 6: Overall daily mean $OP_v$ contribution of the sources to PM$_{10}$, $OP_v^{DTT}$, $OP_v^{AA}$, and $OP_v^{DCFH}$ using MLR analysis in the form of mean and 95% confident interval of the mean (error bar) (n=378 samples).**

The mass contributions of the biomass burning source can be twice as much as that of the primary traffic source, but OP
contributions in terms of $OP_v^{DTT}$ are almost similar. The industrial source also has very minimal contribution in terms of PM$_{10}$
mass, but has relevant contribution to $OP_v$. Moreover, there are sources that contribute to a large extent to the total PM$_{10}$ mass
but barely contribute to the OP, such as the nitrate-rich (all OP assays) and sulfate-rich source (only for $OP_v^{AA}$ and $OP_v^{DCFH}$).
This observed redistribution of source impacts based on $OP_v$ highlights the importance of considering PM redox activity
instead of solely mass concentration (Daellenbach et al., 2020).

Although secondary inorganic sources are commonly associated with low impact on PM toxicity (Cassee et al., 2013;
Daellenbach et al., 2020), the sulfate- and nitrate-rich sources showed contributions to $OP_v^{DTT}$ and $OP_v^{DCFH}$, respectively. Even
with minimal $OP_m$ (see Figure 5), the relevant mass contribution of these sources resulted to relevant contribution to $OP_v$. It
should also be considered that both sulfate- and nitrate-rich sources have been previously associated to anthropogenic SOA
due to phthalic acid contribution in this factor (Borlaza et al., 2020).

Clearly, the $OP_v$ contribution of the biomass burning source is captured by all assays. In fact, in the AA and DCFH assays, the
$OP_v$ contributions are both heavily dominated by the biomass burning source, while the DTT assay showed sensitivity to a
wider range of sources. However, it is important to take into consideration the mechanism at work behind these assays. Both
DTT and AA assays mimic in vivo interactions of redox active components in PM$_{10}$ and biological oxidants representing PM-
induced oxidative stress, while DCFH measures generated particle-bound ROS. Although, these source-specific $OP_v$
contributions provide critical knowledge on the main drivers of $OP_v$, it is difficult to rely on just one measurement (i.e., one
type of assay) without testing its relevance to health outcomes.





### 3.3.4 Seasonal and site-specific differences in OP contribution ($OP_v$) by each PM$_{10}$ source

Clearly, the previous yearly averages mask strong seasonal variabilities as presented in the monthly $OP_v$ contributions of each source (see Figure 7). During colder months, the $OP_v$ of the biomass burning source is present in all assays and especially
prominent in the AA and DCFH assays. During warmer months, the source $OP_v$ contributions varies across different assays. However, the $OP_v$ contributions from the primary traffic source is present throughout the year. Aside from seasonal influences, there are also differences between the sites, that varies according to the assay.

For $OP_v^{DTT}$, there are similarities in the contributions of some sources in the UB and PU sites such as the consistent monthly contribution from the sulfate-rich source and the contributions from the secondary biogenic source during warmer months
highlighting the influence of secondary aerosols in these sites. The UB and UH sites also have similarities in terms of $OP_v^{DTT}$ contributions from the mineral dust source during warmer months and from the nitrate-rich source during the colder months, both of which are sources that can be influenced by road emissions and anthropogenic SOA. This can be explained by the proximity of the UB and UH sites to road ways, where PM$_{10}$ in these sites are more inclined to interact with metals from road dust resuspension and other non-exhaust vehicular emissions than the PU site (discussed in detail in the companion paper
(Borlaza et al., 2020)). Surprisingly, there is also a similarity seen in the UH and PU sites in terms of $OP_v^{DTT}$ contributions from the primary biogenic source during warmer months.

For $OP_v^{AA}$, the contribution from the mineral dust source during warmer months in the UB and UH sites and the contribution from secondary biogenic oxidation source in the PU site were similarly captured. During colder months, biomass burning is dominating in the UB and PU sites, however the UH site exhibited contributions from a variety of sources. There is also a
consistent $OP_v^{AA}$ contribution of aged sea salt in the UB site and the contribution of nitrate-rich and sea/road salt during the colder months in the PU site.

For $OP_v^{DCFH}$, the contributions from the primary traffic source (especially in the UB and PU sites) is much less than the two other assays suggesting weaker sensitivity of DCFH assay to this source. Instead, the contributions from the nitrate-rich source, a source also commonly associated with secondary anthropogenic emissions (Aksoyoglu et al., 2017; Boyd et al., 2017; Faxon
et al., 2018; Pennino et al., 2016; Priestley et al., 2018), is more prominent during the colder months in all sites.

These further highlights not only the importance of PM redox activity over mass concentration, but also the importance of considering the seasonal influence to PM sources that drive the OP of PM. These findings are also consistent with current research underlining that the main sources of OP are those including species mainly originating from anthropogenic emissions (Janssen et al., 2014; Shi et al., 2006; Yang et al., 2015) such as road transport and biomass burning (Boogaard et al., 2012;
Borlaza et al., 2018; Calas et al., 2019; Daellenbach et al., 2020; Daher et al., 2014; Pant et al., 2015; Park et al., 2018; Seo et



al., 2020; Simonetti et al., 2018; Weber et al., 2021) and also site typologies that favour the accumulation of pollutants and photo-active aging (Daellenbach et al., 2020; Janssen et al., 2014; Pietrogrande et al., 2019).



**Figure 7: The monthly mean $OP_v$ contributions of each PM$_{10}$ source in the three urban sites in Grenoble, France for $OP_v^{DTT}$, $OP_v^{AA}$, and $OP_v^{DCFH}$.**

### 3.4 Predicting OP activity from PM$_{10}$ sources using MLP analysis

The residuals between the observed and the MLR-modelled OP could be accounted to atmospheric processes that were not captured as most linear models assume no interaction between independent variables (i.e., multicomponent or multisource interactions). With this in mind, we are inclined to explore another method of predicting OP from PM$_{10}$ sources that hopefully





addresses this limitation. The application of ANN techniques using non-linear functions, such as MLP analysis, is an interesting new approach that accounts for correlation and/or non-linear interactions between independent variables.

### 3.4.1 Optimization of the MLP neural network architecture


A number of MLP architectures (8 architectures in each site (total of 24 MLP models)) were explored to find the optimal neural network in each site by exploring two different activation functions (Hyperbolic tangent (TanH) and Sigmoid), optimization algorithm (Scaled conjugate and Gradient descent), and different learning rates (from 0.2 to 0.6). In the supplementary information (S3), Table S3 shows the performance comparison of all of the MLP models tested. The optimal model was selected based on the lowest RMSE (ideally nearly 0) and highest Pearson correlation coefficient ($r$) (ideally nearly 1). Other model performance measures such as mean absolute error (MAE), mean absolute percentage error (MAPE), and


Spearman rank correlation coefficient ($r_s$) were also explored and lead to relatively similar results.

It is important to note that, although there are other more complex architectures, we limited our tests to a rudimentary MLP architecture that is deemed sufficient and appropriate based on the type of input and output dataset of this study. Clearly, there is room for further exploration in the direction of using MLP for predicting OP from PM sources. To our knowledge, this is the first attempt to use MLP on apportioning OP from PM sources and may serve as a baseline for future applications of MLP


in PM toxicity.

### 3.4.2 Comparison of predictive accuracy between MLP and MLR models

To conduct insightful evaluation of the predictive accuracy of the MLP and MLR models, the model performance measures were calculated as shown in Table 1. The predicted $OP_v^{DTT}$ by the MLP model generally showed lower prediction error (RMSE) than the MLR model for all the sites. Conversely, the model performance measures in $OP_v^{AA}$ and $OP_v^{DCFH}$ were less


straightforward. The predicted $OP_v^{AA}$ showed lower prediction errors for the UB and UH site using MLP models, while lower prediction errors for the PU site using MLR models.

**Table 1: The comparison of predictive accuracy of the observed OP activity between the MLR and MLP models based on root mean square error (RMSE) and Pearson correlation ($r$). Note: RMSE is ideally ~0 (lower RMSE in bold), $r$ is ideally ~1 (higher $r$ in bold).**

| Site | Model | Root mean square error (RMSE) | | | Pearson Correlation coefficient ($r$) | | |
|---|---|---|---|---|---|---|---|
| | | $OP_v^{DTT}$ | $OP_v^{AA}$ | $OP_v^{DCFH}$ | $OP_v^{DTT}$ | $OP_v^{AA}$ | $OP_v^{DCFH}$ |
| Urban background (UB) | MLP | **0.35** | **0.32** | **0.19** | **0.94** | **0.97** | **0.98** |
| | MLR | 0.38 | 0.32 | 0.21 | 0.93 | 0.97 | 0.98 |
| Urban hyper-center (UH) | MLP | **0.54** | **0.50** | **0.30** | **0.88** | **0.94** | **0.95** |
| | MLR | 0.69 | 0.90 | 0.31 | 0.79 | 0.80 | 0.94 |
| Peri-urban (PU) | MLP | **0.58** | 0.44 | 0.32 | **0.75** | 0.97 | 0.96 |
| | MLR | 0.62 | **0.42** | **0.31** | 0.71 | **0.97** | **0.96** |






The temporal distribution of the observed and modelled OP activities for both MLR and MLP models were previously presented in Figure 4. It is interesting to note that even MLP was not able to fully capture some peaks (especially in the warmer months) of the observed $OP_v^{DTT}$. However, the RMSE values using MLP were much lower than MLR, particularly in the UH site where the RMSE was reduced from 0.69 to 0.54, and in the PU site from 0.62 to 0.58. In the UB site, the MLP did not

exceed the performance of MLR by a weighty extent. Nonetheless, the MLP model generally performed better making it a competitive new technique in predicting OP activity even with a rudimentary MLP architecture.

### 3.4.3 The non-linearity of OP contributions of PM$_{10}$ sources based on MLP analysis

With some interactions between PM$_{10}$ sources resulting to synergistic or antagonistic effects on the OP activity, it is deemed essential to look closer into this potential non-linear aspect to understand better the oxidizing capacity of PM$_{10}$ sources. To

demonstrate this non-linearity, the MLP models were applied to dummy datasets leading to source-specific $OP_v$. The total source-specific $OP_v$ ($MLP_{sum}$, see section 2.4.3.2) was compared to the original MLP-modelled $OP_v$ as presented in Figure 8 for the $OP_v^{DTT}$ in the UH site (see supplementary information (S7) for similar figures for the UB and PU sites). The data points below the 1:1 line shows an overall synergistic effect between PM$_{10}$ sources on $OP_v$, while data points above the 1:1 line shows an overall antagonistic effect between PM$_{10}$ sources on $OP_v$.


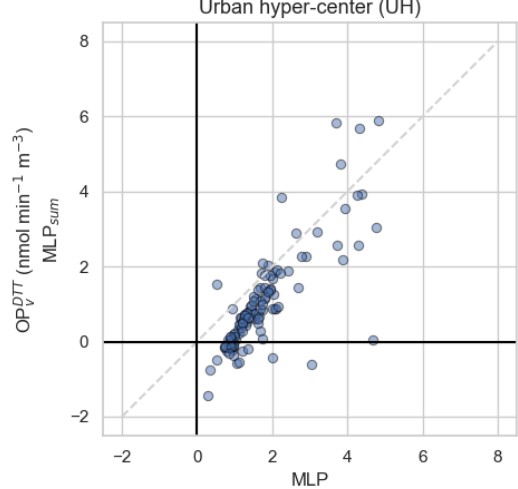

**Figure 8: The comparison of the original modelled $OP_v^{DTT}$ ($MLP$) and the sum of source-specific modelled $OP_v^{DTT}$ activity. Note: Dashed grey line corresponds to the 1:1 line. Data points below the 1:1 line shows an overall synergistic effect between PM$_{10}$ sources on OP activity, above the 1:1 line is otherwise.**

Overall, there is a synergistic effect of PM$_{10}$ sources on $OP_v^{DTT}$ in most days in the UH site. This is also seen in the $OP_v^{AA}$ and $OP_v^{DCFH}$ (see Figure S9 in the supplementary information (S8)). Several studies have reported synergistic effects in OP due to the interaction between metal and organic species (Arangio et al., 2016; Charrier and Anastasio, 2015; Dou et al., 2015; Fang



et al., 2017; Li et al., 2012; Lin and Yu, 2020; Xiong et al., 2017; Yu et al., 2018). The UH site has pertinent contributions coming from the mineral dust source (high in metal species, possibly combined with anthropogenic organics, from road dust resuspension) and primary biogenic source (high in organic species) which could be initiating the synergistic effects (see Figure S7 in the supplementary information (S8)). While there are relevant contributions from biogenic sources in the other two sites, their mineral dust source is not as high as in the UH site (or vice versa). These findings further support the importance of accounting the contribution of biogenic sources as previously reported in other similar studies (Samake et al., 2017; Tuet et al., 2017) as well as the importance of source interactions and dynamics as it could have considerable influence on the OP of $PM_{10}$.

In section 3.4.2, it was presented that MLP offered improvements compared to MLR, based on its much lower prediction errors in the UH site (see Table 1). Indeed, it is possible that MLR had difficulties to generate an accurate OP model for a site that has a highly non-linear behaviour based on the potential synergistic effects between $PM_{10}$ sources. In fact, the lowest prediction error by MLR ($OP_v^{DTT}$ model in the PU site with RMSE=0.21, see Table 1) also showed data points closer to the 1:1 line between the $MLP$ vs $MLP_{sum}$ (see Figure S9 in the supplementary information (S8)) suggesting weaker influence of the synergistic/antagonistic effects between $PM_{10}$ sources. However, the MLP still performed better ($OP_v^{DTT}$ model in the PU site with RMSE=0.19, see Table 1) supporting the flexibility of MLP in both linear and non-linear behaviour of $PM_{10}$ sources compared to MLR.

## 4 Conclusion

This study, together with the findings of its companion paper (Borlaza et al., 2020), have presented an extensive analysis of a city-scale OP and its association to various sources of $PM_{10}$ based on a one-year $PM_{10}$ sampling over different sites in Grenoble (France), with approaches using both linear and non-linear modelling techniques. The main findings of this study are as follows:

- There is a strong seasonality in the observed OP found in all assays used (AA, DTT, and DCFH), with higher OP during colder months and lower OP during warmer months.
- There is a notable spatial difference in OP in a suburban typology against sites closer to the city-center.
- There is an overall agreement (spatiotemporal homogeneity) between the 3 sites in the Grenoble basin, however, there are some influences from local features and site-specific events due to specific sources' contribution.
- The OP of $PM_{10}$ has been successfully attributed to PMF-resolved sources using Multiple Linear Regression analysis with mostly good model fit.
- The sources of OP with highest redox characteristics (i.e., intrinsic OP or $OP_m$) are mainly anthropogenic sources such as industrial, primary traffic, and biomass burning sources. The redox characteristics of commonly unresolved sources in the biogenic fraction were also obtained and such natural sources also contribute to the overall OP during mild seasons.



• There is a redistribution of the impacts in terms of source $OP_v$ contributions compared to mass contributions, highlighting the importance of considering redox activity over mass concentration in Air Quality policies.

• There are seasonal influences on sources contributing to OP. During the colder months, the biomass burning source is typically the strongest contributor to all OP. During the warmer months, there are different sources (mineral dust, primary biogenic, secondary biogenic oxidation) contributing to OP in each site. However, there is a consistent

contribution from the primary traffic source during the overall year.

• Even with a rudimentary design, the Multilayer Perceptron approach successfully modelled OP based on PMF-resolved sources, with some improvements on model performance (lower prediction errors, higher association to observed OP) compared to MLR.

• The MLP also offered improvements especially in sites where there are prominent synergistic and/or antagonistic

effects between $PM_{10}$ sources supporting the capabilities of MLP in capturing non-linearities in OP.

Finally, in this paper, we tested for the very first time the use of neural network analysis to apportion OP sources from $PM_{10}$. We showed that such methodology is at least as robust as the linear classical inversion one and permits an improvement in the OP prediction when local features or non-linear effects occur. This study also demonstrated that enhanced-PMF solution allows to show differences in the spatiotemporal distribution of OP activity, targeting the responsible sources, at a city-scale. These

findings pave the way of establishing exposure in homogenous OP areas.

### Acknowledgements

This work is supported by the French National Research Agency in the framework of the "Investissements d'avenir" program (ANR-15-IDEX-02), for the MobilAir program and ANR Get OP Stand OP (ANR-19-CE34-0002-01). It also received support from the program QAMECS funded by ADEME (convention 1662C0029), and from LCSQA and French Ministry of

Environment for part of the analyses for the Les Frenes site within the CARA program. Chemical analysis on the Air-O-Sol facility at IGE was made possible with the funding of some of the equipment by the Labex OSUG@2020 (ANR10 LABX56). The PhD of SW is funded by ENS Paris. The internship of T Cañete is taking place within the Erasmus exchange program. Finally, the authors would like to kindly thank the dedicated efforts of many people from Atmo-AuRA at the sampling sites, and in the lab at IGE (A Vella, C Vérin, C Voiron, R El Azzouzi, and A Crouzet) for collecting and analysing the samples,

respectively.

### Authors contributions

GU, and JLJ designed the atmospheric chemistry part of the MobilAir and QUAMECS program and the whole Get OP stand OP project. SM and CT supervised the sampling at the 3 sites for Atmo AuRA. AA is involved in the CARA program that

allows the collection of samples from Les Frênes site. SH is developing OP assays for the groupLJSB and SW processed the



data. SW developed some of the tools and ideas for in-depth PMF analysis. LJSB, SW, wrote the paper. JLJ and GU revised the original draft. All authors reviewed and edited the manuscript.

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
