# Peer review of "Disparities in particulate matter $(PM_{10})$ origins and oxidative potential at a city-scale (Grenoble, France) - Part II: Sources of $PM_{10}$ oxidative potential using multiple linear regression analysis and the predictive applicability of multilayer perceptron neural network analysis"

_Atmospheric Chemistry and Physics, 2021_

## Author Comment (AC1)

**Disparities in particulate matter (PM$_{10}$) origins and oxidative potential at a city-scale (Grenoble, France) - Part II: Sources of PM$_{10}$ oxidative potential using multiple linear regression analysis and the predictive applicability of multilayer perceptron neural network analysis**

Authors' response

We would like to thank the referees for their time to evaluate our manuscript and for their positive and constructive feedbacks, which helped improving the quality of the paper. Our point-by-point response to the comments are presented below with the referee comments in black, our answers in red, and changes in the revised version of the manuscript are printed in blue.

Anonymous Referee #1:

The manuscript reports the second part of the results of the chemical analyses of a yearly set of PM10 samples collected at three urban sites in Grenoble (FR). The assessment of the main source contributions performed by positive matrix factorization (PMF) is discussed in the first paper (Borlaza et al., 2020) while here the focus is on oxidative potential (OP) and its relation to the PMF factors identified in the first study. This is carried out using simple multiple linear regression (MLR) analysis as well as by an artificial neural network (ANN) approach: the multilayer perceptron analysis (MLP). This is probably among the first applications of machine learning techniques to the investigation of the chemical characteristics of PM determining its OP. The MLP analysis can account for possible non-linear behaviours of OP due to synergic or antagonistic effects between reactive PM chemical compounds, hence providing a more realistic representation of the way OP is determined by aerosol compounds present in mixtures. I list my major comments below:

The MLR method has a clear advantage on MLP: it enables to assess OP source contributions explicitly (Fig. 5). Therefore, it is mainly by means of MLR that this study addresses the main policy-relevant questions on the sources of aerosol toxicity in this environment. Ideally, the individual sources would carry constant specific intrinsic OP and the different OPv levels between sites would be explained entirely by the spatio-temporal variability of the sources. This is only partly achieved, because undefined "site-specific features" (line 358) remain. Most notably, the specific intrinsic OP of the industrial factor is inconsistent between the PU site and the UB+UH sites for two out of three OP assays. It is therefore unclear on what basis, OP source contributions can be generalized and averaged between the three sites (Fig. 6) to provide a ranking of them. The impression is that the PMF analysis was unable to capture the full sources of OP variability in this environment.

Reply: Thank you very much for this comment. We agree with the reviewer that the MLR method has an advantage by being able to apportion the sources of aerosol toxicity (OP). In this regard, the MLR method does have an advantage on MLP. However, even a rudimentary design of the MLP architecture offered improvements on OP prediction. The successful application of MLP in this study paves the way of using MLP (or other artificial neural network (ANN) techniques) in OP studies. In fact, the use/comparison of different ANN-based methodologies on OP of PM is an on-going study in our group.
This two-part paper elucidates the disparities found in PM sources and OP at a city scale. The city-scale variabilities found were attributed to influences by season-specific sources, site typology, and occurrence of specific local features, as well as assay sensitivity.

In fact, the companion paper (https://doi.org/10.5194/acp-21-5415-2021) dealt with the homogeneity of sources across the city using an advanced method of comparison of the factor chemical profile. The industrial factor was identified as a heterogeneous source. However, it is important to note that the impact of trace metals is inherently variable at this spatial scale. It is also known that emissions from industrial activities are very variable in real life, even from day to day by a single source. The authors have decided to label/identify this factor as "industrial" despite this variability. This is supported by the plots (including Figure 6) which indicates that the standard deviation produced by averaging is acceptable and the sources can still be ranked. For clarity, this is now further highlighted in the paper as follows:

Action: This source has been previously identified as a heterogeneous source in the companion paper. It is important to note that the impact of trace metals, used to identify this source (i.e., As, Cd, Cr, Mn, Mo, Ni, Pb, Zn), is inherently variable at this spatial scale.

Finally, the disparities in sources of OP from different urban site types is one of the key points discussed (see section 3.2). The results suggest that OP source contributions could potentially be generalized on sources that are homogeneous in the city. Otherwise, we cannot generalize OP characteristics of sources with heterogeneous chemical profiles (e.g., industrial, mineral dust factor).

The authors would also like to point out that the PMF methodology, indeed, has its limitations in apportioning PM mass to its sources. However, based on the mass closure in all sites, the reconstructed $PM_{10}$ contributions from all sources and measured $PM_{10}$ concentration indicated very good model results (see Line 319 to 321 in the companion paper) (UB: r=0.99, n=125, $p<0.05$; UH: r=0.99, n=126, $p<0.05$; and PU: r=0.99, n=126, $p<0.05$).

The MLP analysis represents the most innovative aspect of the methodology. However, the results show that the performance of MLP is not really superior to that of MLR in reproducing the observed OP, except for the AA assay at the UH site. The true highlight about MLP stands in its ability to detect non-linear behaviours between chemical compounds. However, such behaviours are not characterized explicitly by the ANN analysis, they remain "hidden" and can only be diagnosed (Fig. 8). The way this is carried in this study (Section 2.4.3.3) is not completely convincing. According to this method, the OP for a given source is estimated as the difference between the modelled OP and the modelled OP obtained on a dummy dataset where the PMF factor for that same source is omitted. However, the PMF factors are not orthogonal, they often exhibit a certain degree of covariance. Even if we remove the nitrate-rich factor, for instance, some features of its time series are still present in the trend of the biomass burning factor. The variability of a given source cannot fully be omitted in the dummy dataset. As a consequence, the source-contribution of OP calculated as a residual (equation 3) risks to be underestimated.

Reply: We thank the reviewer for an in-depth comment on the methodology. Indeed, the PMF factors does exhibit a certain degree of covariance. However, based on the Bootstrap runs (Table S4 in the companion paper supplementary information, https://acp.copernicus.org/articles/21/5415/2021/acp-21-5415-2021-supplement.pdf), most sources were correctly mapped to factors. Only the Sulfate-rich factor had unmapped runs (0.7 mean unmapped runs) with a range of 88 to 99% correct BS runs.

With this information, we assume that the variability of a given source is predominantly captured by the proposed methodology (Equation 3). However, we are aware that there are risks in this estimation just like any other model. To our knowledge, this is the first attempt to computationally demonstrate the non-linear behaviour of OP sources. The authors believe that,
as such, it merits publication in order to open this road to more work, some of it being currently
performed in our group.
Specific comments:
Line 29 (Abstract): "underlining the importance of PM redox activity over mass concentration".
This statement is unclear: is this a claim for PM redox activity being a superior metric respect
to PM mass concentrations? I do not think the Authors would dare to say that. I suggest to
rephrase into something like: "underlining the importance of PM redox activity for the
identification of potential sources of PM toxicity".
Reply: Thank you for this suggestion. This has been revised in the manuscript accordingly.
Action: There is also a clear redistribution of source-specific impacts when using OP instead of
mass concentration, underlining the importance of PM redox activity for the identification of
potential sources of PM toxicity.
Lines 38-39: "explore both the small- and large-scale variabilities of PM pollution accounting
for local variations in different urban environments". Please, rephrase more clearly. What are
the actual spatial scales at issue? What are the "different urban environments"? cities with
different characteristics or different economic districts within a single city?
Reply: The term different "urban environments" referred to here is the "urban typologies". For
clarity, this has been revised in the manuscript as:
Action: The intricate topography and seasonality of particulate air pollution in the city of
Grenoble (France) makes it an ideal location to explore variabilities of PM pollution, while also
accounting for different site typologies within a single medium-sized city (Calas et al., 2019;
Favez et al., 2010; Srivastava et al., 2018; Tomaz et al., 2016, 2017; Weber et al., 2019).
Lines 49 – 50: The definition of oxidative potential (OP) introduced by the Authors suggests
that OP can really traces the ability of aerosol particles to induce oxidative stress in biological
systems. However, the actual link between the OP determined by acellular assays and the ROS
assays employing in vitro system is still currently matter of debate between scientists (e.g., J
Øvrevik, International journal of molecular sciences 20 (19), 4772). The Authors are
encouraged to provide a concise treatment of this fundamental issue.
Reply: We appreciate this comment. Our group has recently published companion papers that
addressed this specific issue and our OP results have been clearly associated with oxidative
stress on lung cells for the same samples (Leni et al., 2020; Daellenbach et al., 2020). However,
for clarity, this sentence has been revised and now reads as:
Action: The oxidative potential (OP) of PM, defined as the capability of PM to generate ROS,
makes an interesting complementary to regulated metrics of ambient PM exposure (Bates et al.,
2019; Daellenbach et al., 2020; Guo et al., 2020; Gurgueira et al., 2002; Park et al., 2018;
Shiraiwa et al., 2017; Valavanidis et al., 2008).

Lines 63 – 64 ("because numerous factors could affect OP"): make such factors explicit.

Reply: Thank you for this comment. This sentence now reads as:

Action: However, a non-linear relationship between redox active components of PM is
generally observed (Arangio et al., 2016; Calas et al., 2017; Charrier and Anastasio, 2015; Li
et al., 2012; Xiong et al., 2017; Yu et al., 2018), hence traditional deterministic models could
be, in some way, limited.

Line 79 ("fine-scale spatiotemporal characteristics"): again, what are the scales of importance
for the present study? If the city scale is the target, 24h-integrated samples collected at three
sampling points is not properly "fine scale". Clearly, there should be more emphasis on the
chemical resolution. Please, explain.

Reply: Thank you for this comment. We agree that the terminology "fine-scale" can be
subjective, but the authors deem this is appropriate based on the configuration (of the Grenoble
basin) and land-use over the metropolitan area. The sampling sites are all within 15 km from
the city-center of Grenoble and each site represents a different urban typology. In fact, UH and
UB are within 4 km of each other. In an ambient PM sampling procedure in a medium-sized
city, we are lucky to be able to sample PM from 3 different types of typologies.

Line 88 ("catch the non-linear pattern of OP"): why non-linear behaviours of OP in this specific
environment are taken for granted?

Reply: Acellular OP assay responses could be dependent on species composition and emissions
source, and may also vary due to multicomponent interactions (e.g., between metals and
organics and/or emission sources) (Arangio et al., 2016; Calas et al., 2017; Charrier and
Anastasio, 2015; Li et al., 2012; Xiong et al., 2017; Yu et al., 2018). The presence of bacteria
has also been reported to influence OP measurements (a reduction up to half of the OP signal)
(Samake et al., 2017). This non-linear behaviour can be taken for granted, especially on studies
focusing only on measured components/species in PM.

Line 133. The term "exposure" can be misleading in this context. Actually, the volume-
normalized OP activity can be related to exposure only upon an assessment of outdoor exposure
itself, which certainly is season-dependent. I would more safely define OPv as the OP carried
by the aerosol expressed in OP units per cubic meter of air.

Reply: We thank the reviewer for this suggestion. This has been revised in the manuscript and
now reads:

Action: The $OP_m$ is the intrinsic OP property of one µg of PM, while $OP_v$ represents the PM-
derived OP per m$^{-3}$ of air.

Figure 7: are source-contributions to OPv calculated using the MLR method? Please specify in
the caption.

Reply: Thank you very much for this clarification. We have updated the figure caption to:

Action: Figure 1: The monthly mean $OP_v$ contributions of each $PM_{10}$ source in the three urban sites in Grenoble, France for $OP_v^{DTT}$, $OP_v^{AA}$, and $OP_v^{DCFH}$ based on MLR analysis.

Anonymous Referee #2:

Bolraza et al. is an interesting manuscript that tested the use of neural network analysis to apportion OP sources in PM10. The manuscript is well written but highly dependent on the companion paper. In reviewer's opinion, there should be one paper by merging this one with the companion paper. In any case, there are many important points in this paper those need to be clarified and addressed before merging it to companion paper or accepting it as an individual publication, depending upon the Editor's decision.

Reply: We appreciate the reviewer's feedback. We agree that this manuscript tends to refer on the companion paper, especially when discussing the sources of PM. The authors deem that it was impractical to merge the two papers, leading to the decision to write a two-part series that are published back-to-back. This way allows for deeper exploration and interpretation of both PM and OP apportionment.
Part 1, which revolved around the source apportionment of PM using fit-for-purpose tracers, already had a lot of interesting results—all of which were worthy of a detailed discussion (https://doi.org/10.5194/acp-21-5415-2021). It is one of the very few attempts to apportion secondary organic aerosol (SOA) sources using more practical and innovative organic tracers. Part 2, which dealt with the sources OP, also contains a great deal of new information, not to mention the first ever to introduce machine learning approach (i.e., Multilayer Perceptron analysis) to improve the prediction of OP from PM sources. This paper (Part 2) could be a standalone publication when one accepts that the PMF results are of grounded result.

Major Comments:

Objectives (Lines 85-88): The objectives of this MS are not satisfactory because this paper can't stand alone. Without companion paper, one can't understand this paper. This is a major draw back. There can be part 1, part 2, etc. of the paper complementing different aspects of a given topic, but each part should also be able stand alone.

Reply: We disagree with the reviewer. This paper mainly revolved around the OP levels of the sources to which one doesn't need the companion paper to understand.
In addition, we have provided sufficient information about the sampling sites and all chemistry analysis done on the previous paper. And finally, we made a synthesis of the methodology used for $PM_{10}$ source apportionment in order to provide a clear view of it to the reader of this single paper.

Section 2.3

Lines 127-129 : Insoluble particles can be a large source of uncertainty, as they are not uniformly mixed in the solution. They can interfere with spectrometric analysis via physical absorbance.

Reply: The extraction procedure in this study is based on Calas et al. (2018), also published by
our group. This procedure has been tested on both soluble and insoluble compounds that are (as
much as possible) within the range of atmospheric concentrations. To avoid the interferences
in the wells by insoluble particles, we subtracted the intrinsic absorbance of all PM extractions
before adding reactants. Also, the particles are extracted in the Gamble solution (an artificial
lining fluid) where we add a surfactant: this was shown to maintain a good dispersion of
particles, leading to homogeneous results (see Calas et al., 2018). This is summarized in Table
S5 of Calas et al. (2018). All analysis was performed in triplicate, with a coefficient of variation
(CV) $\leq 5\%$.

Lines 134-135: This suggests the precision of the measurements. How do you ascertain the
accuracy of the measurements for each assay?

Reply: In every experiment, a positive control 1,4 naphtoquinone and an ambient filter (PM
sampled from the lab roof) were analysed to ensure accuracy of measurements. All analysis
was also performed in triplicate, with a coefficient of variation (CV) $\leq 5\%$.

Lines 144-146: How do you ensure the uniformity of insoluble particles in each well? This
needs to be clarified.

Reply: This has been deeply investigated in Calas et al. (2018). To ensure the uniformity of
insoluble particles, we add a natural lung surfactant (DPPC) in the PM extraction lining fluid
to mimic more closely the contact of PM with lungs and maintaining a homogeneous dispersion
of the particles as shown by the better accuracy of the measurements with DPPC than without
(Calas et al., 2018). Section 2.3 discusses briefly this procedure, but we suggest the readers to
refer to Calas et al. (2018) for more information regarding the extraction procedure.

Lines 155-160: DCFH output is often reported in the form of equivalent H2O2. Here is is
reported as nmol/min/m3. Authors shall provide the details, and also show the linearity in H2O2
formation as a function of time before using this unit.

Reply: Thank you for this comment. In Line 159, we have stated in the methodology that "The
ROS concentration in the sample is calculated in terms of $H_2O_2$ equivalent based on a $H_2O_2$
calibration (100, 200, 300, 400, 500, 1000, and 2000 nmol).". Please take note that the unit for
$OP_v^{DCFH}$ is in terms of $H_2O_2$ equivalent. The calibration curve in this range is linear in every
experiment with $r^2 > 0.96$.

Lines 205-206: In the algorithm, what was criterion of determining the number of neurons in
the hidden layer?

Reply: In Line 206, we mentioned that the number of neurons in the hidden layer was
determined automatically by the estimation algorithm. By implementing MLP in SPPS, there
is an option for an automatic architecture selection. The operator specifies the minimum (n=1)
and maximum (n=50) number of units allowed in the hidden layer, and the automatic architecture selection computes the "best" number of units in the hidden layer. Take note that
"best" refers to the output (i.e., OP prediction) closest to the observed OP activity. In most tests
performed for this study, the number of neurons in the hidden layer are often around 6.
Line 217: What is the rationale behind choosing 80% and 20% only?
Reply: Thank you for this question. This is a common ratio of partition. Some use 75% and
25% for their training and testing sets, respectively. However, general practice will be around
these values. There is no standard way of performing MLP analysis yet. Generally, one should
choose based on *a priori* knowledge and the size of the dataset. A higher percentage for the
testing set could be more suitable to bigger datasets. For this study, we have opted for a general
ratio of partition.
Lines 229-232: Was the output unique for given input parameters? Or, different input
parameters can give same/similar output?
For example: If MLP gives OPv value 'x' for a% of BB, b% of Primary traffic, c% of Mineral
dust, d% of Industries, and so on, then, can i% of BB, j% of Primary traffic, k% of Mineral
dust, l% of Industries, and so on, also give the same output value (x) of OPv? How do you
check whether the ouput is unique or not?
Reply: Every output of the MLP analysis produces a unique result (i.e., predicted OP activity).
They do not have exactly the same values each run, but they are relatively in the same order of
magnitude. We only solely tested on using the PMF-resolved sources in the input layer.
Lines 272-274: What could be the reason for the observed correlations between different assays
when they are known to respond to different species? Can you make an inference that one can
use only a particular assay rather than all the three assays?
Reply: The observed correlations, for example, between DTT and AA assay could be due to
their similar sensitivity to some species. Please see table below for a summary of a few
publications on OP assays and their correlations to chemical species. This table has also been
added in the supplementary information (S9) as Table S4 and mentioned in the main text as:
Action: Table S4 in the supplementary information (S9) summarizes several publications on
OP assays and their correlations to chemical species.
Table S4. Summary of publications relating OP assays to chemical species.

| OP assay | Species driving responses in OP assay | Source |
| --- | --- | --- |
| DTT | soluble nonspecific metals | (Shinyashiki et al., 2009) |
| | soluble copper | (Charrier and Anastasio, 2012, 2015; Charrier et al., 2016; Borlaza et al., 2018; Park et al., 2018; Joo et al., 2018) |
| | soluble manganese | (Charrier and Anastasio, 2012, 2015; Charrier et al., 2016; Borlaza et al., 2018; Park et al., 2018; Joo et al., 2018) |
| | OC (including WSOC and WIOC) | (Cho et al., 2005; Fang et al., 2016; Verma et al., 2012, 2015b; Jeng, 2010; Hu et al., 2008; Verma et al., 2011, 2009; Velali et al., 2016; |

| | | Vreeland et al., 2017; Liu et al., 2014; Borlaza et al., 2018; Park et al., 2018; Joo et al., 2018) |
|---|---|---|
| | PAHs and quinones | (Cho et al., 2005; McWhinney et al., 2013; Chung et al., 2006; Totlandsdal et al., 2015) |
| | HULIS | (Verma et al., 2012, 2015b, a; Dou et al., 2015; Ma et al., 2018) |
| AA | soluble copper | (DiStefano et al., 2009; Fang et al., 2016; Visentin et al., 2016) |
| | total copper | (Janssen et al., 2014; Pant et al., 2015) |
| | total iron | (Janssen et al., 2014; Godri et al., 2010, 2011) |
| | soluble iron | (Koehler et al., 2014) |
| | total lead | (Godri et al., 2010) |
| | total zinc | (Godri et al., 2011) |
| | soluble manganese | (Visentin et al., 2016) |
| | OC | (Calas et al., 2018) |
| DCFH | soluble nonspecific metals | (DiStefano et al., 2009) |
| | soluble copper | (Charrier et al., 2014; Wang et al., 2010) |
| | soluble iron | (Wang et al., 2010) |
| | soluble zinc | (Wang et al., 2010) |
| | Quinones | (Xiong et al., 2017) |

It would be difficult to make an inference on which assay to use without testing the relevance of these metrics towards health data. Our results show that $OP_v^{DTT}$ showed sensitivity to a wider range of sources, whereas $OP_v^{AA}$ address both traffic and biomass burning and $OP_v^{DCFH}$ both showed sensitivity mainly towards biomass burning (section 3.3.3).

There is a need for studies associating OP activity (obtained from various assays) to health data (i.e., health outcomes) before an inference can be attempted. In fact, this is also an on-going task in our group.

Lines 281-283: This is very important point of the paper but not clear at all. Mass-normalised assays obviously depend upon the PM composition and not the PM mass. Different assays respond to different species. The statement written in lines 281-283 is confusing. Please elaborate this sentence in detail.

Reply: The comparison of the two measures ($OP_m$ and $OP_v$) allows us to see its dependency on mass concentration. An *r*-value of 0.76 between variable A and B represents a direct proportionality between two variables. Since, $OP_v$ is calculated by multiplying $OP_m$ by mass concentration, then the linear relationship between the two measures is actually the dependence of both measures to mass concentration—mainly driven by meteorological conditions especially in the Alpine valleys.

Fig. 5: BB is not contributing to OP-DTT as much as it contribute to OP-AA and OP-DCFH. This is unexpected as OP-DTT is most responsive to organics. Please explain.

Reply: Thank you for this comment. We acknowledge the fact that OP from DTT assay has been reported to be responsive/sensitive to organics. However, recent studies have reported that OP from DTT assay is not affected by some metals (specifically iron) like other assays, namely AA and GSH. Because of this, OP from DTT assay may not fully capture ROS generated through Fenton chemistry or even the synergistic effects with regards to •OH generation as reported by Xiong et al. (2017). Similarly, Yu et al. (2018) has reported that soluble manganese showed synergistic effects with quinones on OP from DTT assay, while soluble copper appears to have an antagonistic effect with quinones on the same assay. On the contrary, manganese showed an antagonistic relationship with quinones on •OH generation. Quinones and soluble iron or copper react synergistically to form •OH.

Generally, there is an undeniable interplay between species that needs to be considered as well as the sensitivity of each assay to species. As much as each analysis attempts to fully characterize the chemistry of PM, there can still be many species that are unmeasured but, in fact, plays a role in ROS generation. Hence, reported associations could be due to similarity in variations with PM concentration rather than a significant causal relationship between assays and PM components.

Due to the sensitivity of DTT assay to wider range of compounds, such as organics and metals, that are present in various sources, this lead to a more balanced distribution of OP sources (and so weighting the contribution of biomass burning with regards to other sources) than the other OP assays, such as AA and DCFH.

Lines 345-355: Why industrial (or other) sources are responding differently to OP at different sites? Explain.

Reply: In the companion paper (section 3.5.1), we have presented the metric PD-SID (Pearson distance and standardized identity distance) that measures (dis)similarities of chemical profiles by each source. There are some sources that have been identified as heterogenous sources, including the industrial source. This means that the tracers used to identify the industrial source can be different between the 3 sites in this study. It could also imply that there is a varying origin of this source across the Grenoble basin. Due to this difference, it is expected that the OP contribution of the industrial source can be different as well, after all it is considered a heterogenous source. A similar comment by Referee #1 has also been addressed in Line 51.

Lines 512-514: "Redox characteristics of commonly unresolved sources were obtained" - What does that mean? Elaborate it further.

Reply: We refer to MSA-rich, primary biogenic, and secondary biogenic oxidation factors as the "commonly unresolved sources in the biogenic fraction". This sentence was revised and now reads as:

Action: The redox characteristics of commonly unresolved sources in the biogenic fraction (MSA-rich, primary biogenic, and secondary biogenic oxidation) were also obtained and such natural sources also contribute to the overall OP during mild seasons.

General comment:

How the OP-DTT, OP-AA, and OP-DCFH of PM10 observed over the study regions compare with the other parts of the world? This should be included and discussed.

Reply: The authors deem that this is outside of the scope/goal of this paper. After all, this is not a review paper on OP studies. However, our group has a paper (currently under review process
in ACPD) that tackles the synthesis of OP measurements over many sampling sites in France.

Minor Comments:

Line 15: This is a strong but invalid statement. OP doesn't quantify anti-oxidant imbalance
because as of now there is no assay available which respond to all the redox-active species
present in PM.

Reply: Thank you for this comment. This sentence now reads as:

Action: The oxidative potential (OP) of particulate matter (PM) measures PM capability to
potentially cause anti-oxidant imbalance.

Lines 49-51: Give a proper definition of OP.

Reply: Thank you for this comment. This sentence now reads as:

Action: The oxidative potential (OP) of PM, defined as the capability of PM to generate ROS,
makes an interesting complementary to regulated metrics of ambient PM exposure (Bates et al.,
2019; Daellenbach et al., 2020; Guo et al., 2020; Gurgueira et al., 2002; Park et al., 2018;
Shiraiwa et al., 2017; Valavanidis et al., 2008).

L279: Are these relationships significant? Provide p values of each R.

Reply: We thank the reviewer for this suggestion. However, the authors deem that it is
unnecessary to provide the $p$-values in the main text. Instead, the significance of the correlations
($p \leq 0.01$) obtained were mentioned in the figure captions in the supplementary information.

Action: All correlations are significant at $p \leq 0.01$.

L325-327: What could be the reason that MLR could not capture high OP events?

Reply: A very simple answer is that linear regression analysis can fail at finding relationships
that are non-linear in nature. If a specific variable increases at a rate of the log of another
variable, then linear regression will not describe the relationship well. We can imagine a similar
scenario in high OP events.

**References**

Borlaza, L. J. S., Cosep, E. M. R., Kim, S., Lee, K., Joo, H., Park, M., Bate, D., Cayetano, M.
G., and Park, K.: Oxidative potential of fine ambient particles in various environments,
Environmental Pollution, 243, 1679–1688, https://doi.org/10.1016/j.envpol.2018.09.074, 2018.

Calas, A., Uzu, G., Kelly, F. J., Houdier, S., Martins, J. M. F., Thomas, F., Molton, F., Charron,
A., Dunster, C., Oliete, A., Jacob, V., Besombes, J.-L., Chevrier, F., and Jaffrezo, J.-L.:
Comparison between five acellular oxidative potential measurement assays performed with
detailed chemistry on PM10 samples from the city of Chamonix (France), Atmos. Chem. Phys., 18, 7863–7875, https://doi.org/10.5194/acp-18-7863-2018, 2018.

Charrier, J. G. and Anastasio, C.: On dithiothreitol (DTT) as a measure of oxidative potential for ambient particles: evidence for the importance of soluble transition metals, Atmos. Chem. Phys., 12, 9321–9333, https://doi.org/10.5194/acp-12-9321-2012, 2012.

Charrier, J. G. and Anastasio, C.: Rates of Hydroxyl Radical Production from Transition Metals and Quinones in a Surrogate Lung Fluid, Environ Sci Technol, 49, 9317–9325, https://doi.org/10.1021/acs.est.5b01606, 2015.

Charrier, J. G., McFall, A. S., Richards-Henderson, N. K., and Anastasio, C.: Hydrogen Peroxide Formation in a Surrogate Lung Fluid by Transition Metals and Quinones Present in Particulate Matter, Environ. Sci. Technol., 48, 7010–7017, https://doi.org/10.1021/es501011w, 2014.

Charrier, J. G., McFall, A. S., Vu, K. K.-T., Baroi, J., Olea, C., Hasson, A., and Anastasio, C.: A bias in the "mass-normalized" DTT response – An effect of non-linear concentration-response curves for copper and manganese, Atmospheric Environment, 144, 325–334, https://doi.org/10.1016/j.atmosenv.2016.08.071, 2016.

Cho, A. K., Sioutas, C., Miguel, A. H., Kumagai, Y., Schmitz, D. A., Singh, M., Eiguren-Fernandez, A., and Froines, J. R.: Redox activity of airborne particulate matter at different sites in the Los Angeles Basin, Environmental Research, 99, 40–47, https://doi.org/10.1016/j.envres.2005.01.003, 2005.

Chung, M. Y., Lazaro, R. A., Lim, D., Jackson, J., Lyon, J., Rendulic, D., and Hasson, A. S.: Aerosol-borne quinones and reactive oxygen species generation by particulate matter extracts, Environ Sci Technol, 40, 4880–4886, https://doi.org/10.1021/es0515957, 2006.

Daellenbach, K. R., Uzu, G., Jiang, J., Cassagnes, L.-E., Leni, Z., Vlachou, A., Stefenelli, G., Canonaco, F., Weber, S., Segers, A., Kuenen, J. J. P., Schaap, M., Favez, O., Albinet, A., Aksoyoglu, S., Dommen, J., Baltensperger, U., Geiser, M., El Haddad, I., Jaffrezo, J.-L., and Prévôt, A. S. H.: Sources of particulate-matter air pollution and its oxidative potential in Europe, Nature, 587, 414–419, https://doi.org/10.1038/s41586-020-2902-8, 2020.

DiStefano, E., Eiguren-Fernandez, A., Delfino, R. J., Sioutas, C., Froines, J. R., and Cho, A. K.: Determination of metal-based hydroxyl radical generating capacity of ambient and diesel exhaust particles, Inhal Toxicol, 21, 731–738, https://doi.org/10.1080/08958370802491433, 2009.

Dou, J., Lin, P., Kuang, B.-Y., and Yu, J. Z.: Reactive Oxygen Species Production Mediated by Humic-like Substances in Atmospheric Aerosols: Enhancement Effects by Pyridine, Imidazole, and Their Derivatives, Environ. Sci. Technol., 49, 6457–6465, https://doi.org/10.1021/es5059378, 2015.

Fang, T., Verma, V., Bates, J. T., Abrams, J., Klein, M., Strickland, M. J., Sarnat, S. E., Chang, H. H., Mulholland, J. A., Tolbert, P. E., Russell, A. G., and Weber, R. J.: Oxidative potential of ambient water-soluble $PM_{2.5}$ in the southeastern United States: contrasts in sources and health associations between ascorbic acid (AA) and dithiothreitol (DTT) assays, Atmos. Chem. Phys., 16, 3865–3879, https://doi.org/10.5194/acp-16-3865-2016, 2016.

Godri, K. J., Duggan, S. T., Fuller, G. W., Baker, T., Green, D., Kelly, F. J., and Mudway, I. S.: Particulate matter oxidative potential from waste transfer station activity, Environ Health Perspect, 118, 493–498, https://doi.org/10.1289/ehp.0901303, 2010.

Godri, K. J., Harrison, R. M., Evans, T., Baker, T., Dunster, C., Mudway, I. S., and Kelly, F.
J.: Increased oxidative burden associated with traffic component of ambient particulate matter
at roadside and urban background schools sites in London, PLoS One, 6, e21961,
https://doi.org/10.1371/journal.pone.0021961, 2011.

Hu, S., Polidori, A., Arhami, M., Shafer, M. M., Schauer, J. J., Cho, A., and Sioutas, C.: Redox
activity and chemical speciation of size fractioned PM in the communities of the Los Angeles
– Long Beach Harbor, https://doi.org/10.5194/acpd-8-11643-2008, 2008.

Janssen, N. A. H., Yang, A., Strak, M., Steenhof, M., Hellack, B., Gerlofs-Nijland, M. E.,
Kuhlbusch, T., Kelly, F., Harrison, R., Brunekreef, B., Hoek, G., and Cassee, F.: Oxidative
potential of particulate matter collected at sites with different source characteristics, Science of
The Total Environment, 472, 572–581, https://doi.org/10.1016/j.scitotenv.2013.11.099, 2014.

Jeng, H. A.: Chemical composition of ambient particulate matter and redox activity, Environ
Monit Assess, 169, 597–606, https://doi.org/10.1007/s10661-009-1199-8, 2010.

Joo, H. S., Batmunkh, T., Borlaza, L. J. S., Park, M., Lee, K. Y., Lee, J. Y., Chang, Y. W., and
Park, K.: Physicochemical properties and oxidative potential of fine particles produced from
coal combustion, Aerosol Science and Technology, 52, 1134–1144,
https://doi.org/10.1080/02786826.2018.1501152, 2018.

Koehler, K., Shapiro, J., Sameenoi, Y., Henry, C., and Volckens, J.: LABORATORY
EVALUATION OF A MICROFLUIDIC ELECTROCHEMICAL SENSOR FOR AEROSOL
OXIDATIVE LOAD, Aerosol Sci Technol, 48, 489–497,
https://doi.org/10.1080/02786826.2014.891722, 2014.

Leni, Z., Cassagnes, L. E., Daellenbach, K. R., El Haddad, I., Vlachou, A., Uzu, G., Prévôt, A.
S. H., Jaffrezo, J.-L., Baumlin, N., Salathe, M., Baltensperger, U., Dommen, J., and Geiser, M.:
Oxidative stress-induced inflammation in susceptible airways by anthropogenic aerosol, PLoS
ONE, 15, e0233425, https://doi.org/10.1371/journal.pone.0233425, 2020.

Liu, Q., Zhang, Y., Liu, Y., and Zhang, M.: Characterization of springtime airborne particulate
matter-bound reactive oxygen species in Beijing, Environ Sci Pollut Res, 21, 9325–9333,
https://doi.org/10.1007/s11356-014-2843-6, 2014.

Ma, Y., Cheng, Y., Qiu, X., Cao, G., Fang, Y., Wang, J., Zhu, T., Yu, J., and Hu, D.: Sources
and oxidative potential of water-soluble humic-like substances
(HULIS$_{WS}$) in fine particulate matter
(PM$_{2.5}$) in Beijing, Atmos. Chem. Phys., 18, 5607–5617,
https://doi.org/10.5194/acp-18-5607-2018, 2018.

McWhinney, R. D., Badali, K., Liggio, J., Li, S.-M., and Abbatt, J. P. D.: Filterable redox
cycling activity: a comparison between diesel exhaust particles and secondary organic aerosol
constituents, Environ Sci Technol, 47, 3362–3369, https://doi.org/10.1021/es304676x, 2013.

Pant, P., Baker, S. J., Shukla, A., Maikawa, C., Godri Pollitt, K. J., and Harrison, R. M.: The
PM 10 fraction of road dust in the UK and India: Characterization, source profiles and oxidative
potential, Science of The Total Environment, 530–531, 445–452,
https://doi.org/10.1016/j.scitotenv.2015.05.084, 2015.

Park, M., Joo, H. S., Lee, K., Jang, M., Kim, S. D., Kim, I., Borlaza, L. J. S., Lim, H., Shin, H.,
Chung, K. H., Choi, Y.-H., Park, S. G., Bae, M.-S., Lee, J., Song, H., and Park, K.: Differential
toxicities of fine particulate matters from various sources, Sci Rep, 8, 17007,
https://doi.org/10.1038/s41598-018-35398-0, 2018.

Samake, A., Uzu, G., Martins, J. M. F., Calas, A., Vince, E., Parat, S., and Jaffrezo, J. L.: The
unexpected role of bioaerosols in the Oxidative Potential of PM, 7, 10978,
https://doi.org/10.1038/s41598-017-11178-0, 2017.

Shinyashiki, M., Eiguren-Fernandez, A., Schmitz, D. A., Di Stefano, E., Li, N., Linak, W. P.,
Cho, S.-H., Froines, J. R., and Cho, A. K.: Electrophilic and redox properties of diesel exhaust
particles, Environ Res, 109, 239–244, https://doi.org/10.1016/j.envres.2008.12.008, 2009.

Totlandsdal, A. I., Låg, M., Lilleaas, E., Cassee, F., and Schwarze, P.: Differential
proinflammatory responses induced by diesel exhaust particles with contrasting PAH and metal
content, Environ Toxicol, 30, 188–196, https://doi.org/10.1002/tox.21884, 2015.

Velali, E., Papachristou, E., Pantazaki, A., Choli-Papadopoulou, T., Planou, S., Kouras, A.,
Manoli, E., Besis, A., Voutsa, D., and Samara, C.: Redox activity and in vitro bioactivity of the
water-soluble fraction of urban particulate matter in relation to particle size and chemical
composition, Environmental Pollution, 208, 774–786,
https://doi.org/10.1016/j.envpol.2015.10.058, 2016.

Verma, V., Ning, Z., Cho, A. K., Schauer, J. J., Shafer, M. M., and Sioutas, C.: Redox activity
of urban quasi-ultrafine particles from primary and secondary sources, Atmospheric
Environment, 43, 6360–6368, https://doi.org/10.1016/j.atmosenv.2009.09.019, 2009.

Verma, V., Pakbin, P., Cheung, K. L., Cho, A. K., Schauer, J. J., Shafer, M. M., Kleinman, M.
T., and Sioutas, C.: Physicochemical and oxidative characteristics of semi-volatile components
of quasi-ultrafine particles in an urban atmosphere, Atmospheric Environment, 45, 1025–1033,
https://doi.org/10.1016/j.atmosenv.2010.10.044, 2011.

Verma, V., Rico-Martinez, R., Kotra, N., King, L., Liu, J., Snell, T. W., and Weber, R. J.:
Contribution of Water-Soluble and Insoluble Components and Their Hydrophobic/Hydrophilic
Subfractions to the Reactive Oxygen Species-Generating Potential of Fine Ambient Aerosols,
Environ. Sci. Technol., 46, 11384–11392, https://doi.org/10.1021/es302484r, 2012.

Verma, V., Wang, Y., El-Afifi, R., Fang, T., Rowland, J., Russell, A. G., and Weber, R. J.:
Fractionating ambient humic-like substances (HULIS) for their reactive oxygen species activity
– Assessing the importance of quinones and atmospheric aging, Atmospheric Environment,
120, 351–359, https://doi.org/10.1016/j.atmosenv.2015.09.010, 2015a.

Verma, V., Fang, T., Xu, L., Peltier, R. E., Russell, A. G., Ng, N. L., and Weber, R. J.: Organic
Aerosols Associated with the Generation of Reactive Oxygen Species (ROS) by Water-Soluble
PM $_{2.5}$, Environ. Sci. Technol., 49, 4646–4656, https://doi.org/10.1021/es505577w, 2015b.

Visentin, M., Pagnoni, A., Sarti, E., and Pietrogrande, M. C.: Urban PM2.5 oxidative potential:
Importance of chemical species and comparison of two spectrophotometric cell-free assays,
Environmental Pollution, 219, 72–79, https://doi.org/10.1016/j.envpol.2016.09.047, 2016.

Vreeland, H., Weber, R., Bergin, M., Greenwald, R., Golan, R., Russell, A. G., Verma, V., and
Sarnat, J. A.: Oxidative potential of PM 2.5 during Atlanta rush hour: Measurements of in-
vehicle dithiothreitol (DTT) activity, Atmospheric Environment, 165, 169–178,
https://doi.org/10.1016/j.atmosenv.2017.06.044, 2017.

Wang, Y., Arellanes, C., Curtis, D. B., and Paulson, S. E.: Probing the source of hydrogen
peroxide associated with coarse mode aerosol particles in southern California, Environ Sci
Technol, 44, 4070–4075, https://doi.org/10.1021/es100593k, 2010.

Xiong, Q., Yu, H., Wang, R., Wei, J., and Verma, V.: Rethinking Dithiothreitol-Based
Particulate Matter Oxidative Potential: Measuring Dithiothreitol Consumption versus Reactive
Oxygen Species Generation, Environ. Sci. Technol., 51, 6507–6514,
https://doi.org/10.1021/acs.est.7b01272, 2017.

Yu, H., Wei, J., Cheng, Y., Subedi, K., and Verma, V.: Synergistic and Antagonistic
Interactions among the Particulate Matter Components in Generating Reactive Oxygen Species
Based on the Dithiothreitol Assay, Environ. Sci. Technol., 52, 2261–2270,
https://doi.org/10.1021/acs.est.7b04261, 2018.

---

## Referee Report (RR1)

Authors have answered/addressed many of my comments; however, a few things should have been added to the revised MS for the better clarity for readers. I suggest addressing the following comments made on author's responses **(text in bold green)**.

My comment: Lines 127-129: Insoluble particles can be a large source of uncertainty, as they are not uniformly mixed in the solution. They can interfere with spectrometric analysis via physical absorbance.

Reply: The extraction procedure in this study is based on Calas et al. (2018), also published by our group. This procedure has been tested on both soluble and insoluble compounds that are (as much as possible) within the range of atmospheric concentrations. To avoid the interferences in the wells by insoluble particles, we subtracted the intrinsic absorbance of all PM extractions before adding reactants. Also, the particles are extracted in the Gamble solution (an artificial lining fluid) where we add a surfactant: this was shown to maintain a good dispersion of particles, leading to homogeneous results (see Calas et al., 2018). This is summarized in Table S5 of Calas et al. (2018). All analysis was performed in triplicate, with a coefficient of variation (CV) ≤ 5%.

**Information about avoiding interferences from insoluble particles is not given in the methodology section. Brief methodology shall be clearly given in this MS, and for details it is fine to give the reference. Please include.**

My comment: Lines 134-135: This suggests the precision of the measurements. How do you ascertain the accuracy of the measurements for each assay?

Reply: In every experiment, a positive control 1,4 naphtoquinone and an ambient filter (PM sampled from the lab roof) were analysed to ensure accuracy of measurements. All analysis was also performed in triplicate, with a coefficient of variation (CV) ≤ 5%.

**This answer is not clear. Was 1,4 nathoquinone used for all the three assays? I don't think all the three assays respond to this chemical. Please explain. How the ambient PM filter was used for the accuracy of all the three assays? It can only be used for the consistency (precision), which is already shown by CV of each analysis. Authors should provide the output of this accuracy experiment for each assay in the main MS. This information will also be very useful for readers.**

My comment: Lines 281-283: This is very important point of the paper but not clear at all. Mass-normalised  assays obviously depend upon the PM composition and not the PM mass. Different assays respond to different species. The statement written in lines 281-283 is confusing. Please elaborate this sentence in detail.

Reply: The comparison of the two measures ($OPm$ and $OPv$) allows us to see its dependency on mass concentration. An $r$-value of 0.76 between variable A and B represents a direct proportionality between two variables. Since, $OPv$ is calculated by multiplying $OPm$ by mass concentration, then the linear relationship between the two measures is actually the dependence of both measures to mass concentration—mainly driven by meteorological conditions especially in the Alpine valleys.

**This discussion is related to Fig. S3 where OPm vs OPv are plotted for all the three assays. The slope of this plot would be the inverse of PM mass. The reason of this plot is still not clear. One should plot OPv vs PM mass conc. The slope of this plot would be average OPm. If the correlation between OPv and PM mass is very strong, it would reflect that the intrinsic OP of PM is uniform over the study site. If there is a poor correlation, intrinsic OP of PM is expected to be variable due to various reasons. I suggest to plot the Fig.S3 again in the recommended form and discuss.**

My comment: Fig. 5: BB is not contributing to OP-DTT as much as it contribute to OP-AA and OP-DCFH. This is unexpected as OP-DTT is most responsive to organics. Please explain.

Reply: Thank you for this comment. We acknowledge the fact that OP from DTT assay has been reported to be responsive/sensitive to organics. However, recent studies have reported that OP from DTT assay is not affected by some metals (specifically iron) like other assays, namely AA and GSH. Because of this, OP from DTT assay may not fully capture ROS generated through Fenton chemistry or even the synergistic effects with regards to •OH generation as reported by Xiong et al. (2017). Similarly, Yu et al. (2018) has reported that soluble manganese showed synergistic effects with quinones on OP from DTT assay, while soluble copper appears to have an antagonistic effect with quinones on the same assay. On the contrary, manganese showed an antagonistic relationship with quinones on •OH generation. Quinones and soluble iron or copper react synergistically to form •OH.
Generally, there is an undeniable interplay between species that needs to be considered as well as the sensitivity of each assay to species. As much as each analysis attempts to fully characterize the chemistry of PM, there can still be many species that are unmeasured but, in fact, plays a role in ROS generation. Hence, reported associations could be due to similarity in variations with PM concentration rather than a significant causal relationship between assays and PM components. Due to the sensitivity of DTT assay to wider range of compounds, such as organics and metals, that are present in various sources, this lead to a more balanced distribution of OP sources (and so weighting the contribution of biomass burning with regards to other sources) than the other OP assays, such as AA and DCFH.

**This discussion shall be appropriately included in the revised MS.**

My comment: Why industrial (or other) sources are responding differently to OP at different sites? Explain.

Reply: In the companion paper (section 3.5.1), we have presented the metric PD-SID (Pearson distance and standardized identity distance) that measures (dis)similarities of chemical profiles by each source. There are some sources that have been identified as heterogenous sources, including the industrial source. This means that the tracers used to identify the industrial source can be different between the 3 sites in this study. It could also imply that there is a varying origin of this source across the Grenoble basin. Due to this difference, it is expected that the OP contribution of the industrial source can be different as well, after all it is considered a heterogenous source. A similar comment by Referee #1 has also been addressed in Line 51.

**Use of multiple tracers for Industrial source is confusing because different tracers respond to OP assays differently. Authors can split this Industrial source in different subsets using their unique proxy. In present form, it is very confusing for the readers.**

My comment: How the OP-DTT, OP-AA, and OP-DCFH of PM10 observed over the study regions compare with the other parts of the world? This should be included and discussed.

Reply: The authors deem that this is outside of the scope/goal of this paper. After all, this is not a review paper on OP studies. However, our group has a paper (currently under review process in ACPD) that tackles the synthesis of OP measurements over many sampling sites in France.

**This is not a correct thinking. Authors have reported the values of three assays over three closely located sites. It will be meaningful to add a paragraph (with a Table) on how the measured OP values compare with some other sites of the world with similar (or different composition). Add some discussion on this comparison. It is obvious that this MS is not a review article. But for readers, it will be useful to see some discussion on 'comparison'.**

My comment: Lines 49-51: Give a proper definition of OP.

Reply: Thank you for this comment. This sentence now reads as:
Action: The oxidative potential (OP) of PM, defined as the capability of PM to generate ROS, makes an interesting complementary to regulated metrics of ambient PM exposure (Bates et al., 2019; Daellenbach et al., 2020; Guo et al., 2020; Gurgueira et al., 2002; Park et al., 2018; Shiraiwa et al., 2017; Valavanidis et al., 2008).

**It should be -
........, defined as the capability of PM to generate ROS/deplete anti-oxidants, ....**

---

## Author Response (AR2)

**Disparities in particulate matter (PM$_{10}$) origins and oxidative potential at a city-scale (Grenoble, France) - Part II: Sources of PM$_{10}$ oxidative potential using multiple linear regression analysis and the predictive applicability of multilayer perceptron neural network analysis**

Authors' response

We would like to thank the referee for their time to re-evaluate the revised manuscript. We appreciate the efforts made to further improve the manuscript. Our point-by-point response to the extra comments are presented below in **bold red**.

Lines 127-129 : Insoluble particles can be a large source of uncertainty, as they are not uniformly mixed in the solution. They can interfere with spectrometric analysis via physical absorbance.

Reply: The extraction procedure in this study is based on Calas et al. (2018), also published by our group. This procedure has been tested on both soluble and insoluble compounds that are (as much as possible) within the range of atmospheric concentrations. To avoid the interferences in the wells by insoluble particles, we subtracted the intrinsic absorbance of all PM extractions before adding reactants. Also, the particles are extracted in the Gamble solution (an artificial lining fluid) where we add a surfactant: this was shown to maintain a good dispersion of particles, leading to homogeneous results (see Calas et al., 2018). This is summarized in Table S5 of Calas et al. (2018). All analysis was performed in triplicate, with a coefficient of variation (CV) $\leq 5\%$.

**Information about avoiding interferences from insoluble particles is not given in the methodology section. Brief methodology shall be clearly given in this MS, and for details it is fine to give the reference. Please include.**

**Reply: To address this comment, we provided additional information in the manuscript that reads:**

**"To avoid the interferences in the wells by insoluble particles, we subtracted the intrinsic absorbance of all PM extractions before adding the reactants. This procedure has been tested on both soluble and insoluble compounds that are likely within the range of atmospheric concentrations. The results have confirmed good dispersion of particles, leading to homogeneous results. A more detailed report is available in Calas et al. (2018)."**

Lines 134-135: This suggests the precision of the measurements. How do you ascertain the accuracy of the measurements for each assay?

Reply: In every experiment, a positive control 1,4 naphtoquinone and an ambient filter (PM sampled from the lab roof) were analysed to ensure accuracy of measurements. All analysis was also performed in triplicate, with a coefficient of variation (CV) $\leq 5\%$.

**This answer is not clear. Was 1,4 nathoquinone used for all the three assays? I don't think all the three assays respond to this chemical. Please explain. How the ambient PM filter was used for the accuracy of all the three assays? It can only be used for the consistency (precision), which is already shown by CV of each analysis. Authors should provide the**

output of this accuracy experiment for each assay in the main MS. This information will also be very useful for readers.

Reply: We thank the reviewer for this comment. To further clarify, in every experiment, both positive controls and an ambient filter were analysed to ensure stability of the OP analysis. The ambient filter sampled from the lab roof has a known and constant expected OP value. The ambient filters were analysed to ensure precision of measurements.
Indeed, 1,4-naphthoquinone (1,4-NQ) was used for positive control tests for DTT and AA assay. Particularly, a 40 µL of 24.7µM stock solution was used for DTT assay and a 80 µL of 24.7µM 1,4-NQ solution for AA assay (Calas et al., 2018, 2017). Finally, we used a 100 nM $H_2O_2$ for DCFH assay. The measurement quality was estimated by calculating the coefficient of variation (CV%) of the positive controls, all CVs were <3% for the 3 assays. These are now added in the manuscript as:
"For positive control tests, the 1,4-naphthoquinone (1,4-NQ) was used for both DTT and AA assays. Particularly, a 40 µl of 24.7 µM stock solution was used for DTT assay and an 80 µl of 24.7 µM 1,4-NQ solution for AA assay (Calas et al., 2017, 2018). A 100 nM $H_2O_2$ was used for DCFH assay. The measurement quality was estimated by calculating the coefficient of variation (CV) of the positive controls, all CVs were <3% for the 3 assays. Additionally, an ambient filter collected from the lab roof, with a known and constant expected OP value, was analysed to ensure precision of OP measurements."

Lines 281-283: This is very important point of the paper but not clear at all. Mass-normalised assays obviously depend upon the PM composition and not the PM mass. Different assays respond to different species. The statement written in lines 281-283 is confusing. Please elaborate this sentence in detail.

Reply: The comparison of the two measures ($OP_m$ and $OP_v$) allows us to see its dependency on mass concentration. An $r$-value of 0.76 between variable A and B represents a direct proportionality between two variables. Since, $OP_v$ is calculated by multiplying $OP_m$ by mass concentration, then the linear relationship between the two measures is actually the dependence of both measures to mass concentration—mainly driven by meteorological conditions especially in the Alpine valleys.

This discussion is related to Fig. S3 where OPm vs OPv are plotted for all the three assays. The slope of this plot would be the inverse of PM mass. The reason of this plot is still not clear. One should plot OPv vs PM mass conc. The slope of this plot would be average OPm. If the correlation between OPv and PM mass is very strong, it would reflect that the intrinsic OP of PM is uniform over the study site. If there is a poor correlation, intrinsic OP of PM is expected to be variable due to various reasons. I suggest to plot the Fig.S3 again in the recommended form and discuss.

Reply: The slope of the plot is in terms of PM mass concentration. Please be reminded of the units of the two measures ($OP_m$ and $OP_v$), discussed in section 2.3. The $OP_m$ is in nmol min$^{-1}$ µg$^{-1}$, while $OP_v$ is in nmol min$^{-1}$ m$^{-3}$.

Fig. 5: BB is not contributing to OP-DTT as much as it contribute to OP-AA and OP-DCFH. This is unexpected as OP-DTT is most responsive to organics. Please explain.

Reply: Thank you for this comment. We acknowledge the fact that OP from DTT assay has

been reported to be responsive/sensitive to organics. However, recent studies have reported that OP from DTT assay is not affected by some metals (specifically iron) like other assays, namely AA and GSH. Because of this, OP from DTT assay may not fully capture ROS generated through Fenton chemistry or even the synergistic effects with regards to •OH generation as reported by Xiong et al. (2017). Similarly, Yu et al. (2018) has reported that soluble manganese showed synergistic effects with quinones on OP from DTT assay, while soluble copper appears to have an antagonistic effect with quinones on the same assay. On the contrary, manganese showed an antagonistic relationship with quinones on •OH generation. Quinones and soluble iron or copper react synergistically to form •OH.

Generally, there is an undeniable interplay between species that needs to be considered as well as the sensitivity of each assay to species. As much as each analysis attempts to fully characterize the chemistry of PM, there can still be many species that are unmeasured but, in fact, plays a role in ROS generation. Hence, reported associations could be due to similarity in variations with PM concentration rather than a significant causal relationship between assays and PM components.

Due to the sensitivity of DTT assay to wider range of compounds, such as organics and metals, that are present in various sources, this lead to a more balanced distribution of OP sources (and so weighting the contribution of biomass burning with regards to other sources) than the other OP assays, such as AA and DCFH.

**This discussion shall be appropriately included in the revised MS.**

**Reply: We have included this discussion in the manuscript :**

**"It is also interesting that biomass burning appears to be contributing less to $OP_m$ in the DTT assay compared to both the AA and DCFH assays. We acknowledge the fact that OP from DTT assay has been reported to be responsive/sensitive to organics making this quite intriguing. However, recent studies have reported that OP from DTT assay could be unreactive to some metal species (specifically iron) unlike other assays, namely AA and glutathione (GSH). Hence, OP measured using DTT assay may not completely capture ROS from Fenton chemistry or even the synergistic effects with regards to hydroxyl radical (•OH) generation as reported by Xiong et al. (2017). Similarly, Yu et al. (2018) has reported that soluble manganese showed synergistic effects with quinones, while an antagonistic effect between soluble copper and quinones. Generally, there is an undeniable interplay between species that needs to be considered as well as the sensitivity of each assay to species. As much as each analysis attempts to fully characterize the chemistry of PM, there can still be species that are unmeasured but, in fact, play a role in ROS generation. Hence, reported associations could be due to similarity in variations with PM concentration rather than a significant causal relationship between assays and PM components. Nevertheless, the sensitivity of DTT assay to a wider range of compounds that are present in various sources, lead to a more balanced distribution of OP sources (and so weighting the contribution of biomass burning with regards to other sources) than the other OP assays, such as AA and DCFH."**

Lines 345-355: Why industrial (or other) sources are responding differently to OP at different sites? Explain.

Reply: In the companion paper (section 3.5.1), we have presented the metric PD-SID (Pearson

distance and standardized identity distance) that measures (dis)similarities of chemical profiles by each source. There are some sources that have been identified as heterogenous sources, including the industrial source. This means that the tracers used to identify the industrial source can be different between the 3 sites in this study. It could also imply that there is a varying origin of this source across the Grenoble basin. Due to this difference, it is expected that the OP contribution of the industrial source can be different as well, after all it is considered a heterogenous source. A similar comment by Referee #1 has also been addressed in Line 51.

**Use of multiple tracers for Industrial source is confusing because different tracers respond to OP assays differently. Authors can split this Industrial source in different subsets using their unique proxy. In present form, it is very confusing for the readers.**

**Reply: The industrial factor is generally identified by high loadings of specific metal species (Figure S3.10 in the companion paper). Although, there is a difference in the chemical profile, the metal species used are all usual tracers of industrial-related sources. The authors deem that it is unnecessary to sub-categorize the industrial sources further.**

General comment:

How the OP-DTT, OP-AA, and OP-DCFH of PM10 observed over the study regions compare with the other parts of the world? This should be included and discussed.

Reply: The authors deem that this is outside of the scope/goal of this paper. After all, this is not a review paper on OP studies. However, our group has a paper (currently under review process in ACPD) that tackles the synthesis of OP measurements over many sampling sites in France.

**This is not a correct thinking. Authors have reported the values of three assays over three closely located sites. It will be meaningful to add a paragraph (with a Table) on how the measured OP values compare with some other sites of the world with similar (or different composition). Add some discussion on this comparison. It is obvious that this MS is not a review article. But for readers, it will be useful to see some discussion on 'comparison'.**

**Reply: Thank you for this comment. We understand the interest of the reviewer on a global comparison of OP levels. We agree it would provide useful information to readers. However, the objective of the manuscript is to investigate the OP variability within a medium-sized urban area and the corresponding influence in terms of contributions of the emissions sources to OP.**
**A global comparison of OP levels will lead towards a discussion tackling the difference in OP protocols across the world-- a topic that deserves a publication on its own. For example, the PM extraction methods could vary by solvent (water, organic, surrogate lung fluid) and conditions (i.e., iso-mass vs non iso-mass). There are also differences in the filter types (Teflon, quartz, zeflour) and sampling procedures (PM size, sampling duration). There also varying methods of calculating OP activity (% depletion, anti-oxidant consumption). These are variables to consider on top of the variabilities brought about by different a-cellular assays (DTT, AA, ESR, DCFH, GSH to name a few).**
**We understand the immense importance of a standard method for OP analysis to facilitate inter-study comparisons across the world. In fact, our group has a paper in review (Weber et al. (2021), https://acp.copernicus.org/preprints/acp-2021-77/acp-2021-77.pdf) that presents a national synthesis in France that could pave way towards inter-study**

**comparisons in the future. To further clarify, we have added the sentence below in the manuscript:**

**"The range of the OP measurements in Grenoble are well within the range of measurements in France (Calas et al., 2018, 2019b; Weber et al., 2021, 2018)."**

Lines 49-51: Give a proper definition of OP.

Reply: Thank you for this comment. This sentence now reads as:

Action: The oxidative potential (OP) of PM, defined as the capability of PM to generate ROS, makes an interesting complementary to regulated metrics of ambient PM exposure (Bates et al., 2019; Daellenbach et al., 2020; Guo et al., 2020; Gurgueira et al., 2002; Park et al., 2018; Shiraiwa et al., 2017; Valavanidis et al., 2008).

**It should be -**
**........, defined as the capability of PM to generate ROS/deplete anti-oxidants, ....**

**Reply: Thank you for this comment. This sentence now reads as:**

**"The oxidative potential (OP) of PM, defined as the capability of PM to generate ROS/deplete anti-oxidants, makes an interesting complementary to regulated metrics of ambient PM exposure (Bates et al., 2019; Daellenbach et al., 2020; Guo et al., 2020; Gurgueira et al., 2002; Park et al., 2018; Shiraiwa et al., 2017; Valavanidis et al., 2008)."**